# 2D reflection seismic surveys to delineate manganese mineralisation beneath the thick Kalahari and Karoo cover in the Griqualand West Basin, South Africa

Mpofana Sihoyiya[1], Musa S. D. Manzi[1], Ian James[1], Michael Westgate[1]

[1]School of Geosciences, University of the Witwatersrand, Johannesburg, Private Bag, Wits, 2050, Republic of South Africa.

*Correspondence to*: Mpofana Sihoyiya (mpofana.sihoyiya@wits.ac.za)

**Abstract.** The Kalahari Manganese Field (KMF) in the Northern Cape Province of South Africa hosts some of the world's richest manganese deposits, largely concealed beneath thick Cretaceous to Cenozoic Kalahari Group and Karoo Supergroup sediments. To improve imaging of the concealed Transvaal Supergroup strata, a high-resolution 2D reflection seismic survey
was conducted in November 2023 across the Severn farm area. The survey comprised five profiles totalling 18.9 km, acquired using 5 Hz 1C geophones connected to wireless nodes, enabling effective burial beneath loose aeolian sand for improved coupling. A compact 500 kg drop hammer, mounted on a Bobcat, served as the seismic source, offering excellent manoeuvrability across challenging sandy terrain. Shot spacing was 10 m, with four vertical stacks per shot to enhance signal-to-noise ratio. Refraction tomography using first-break travel times provided near-surface P-wave velocity models, revealing
variable Kalahari sediment thicknesses ranging from 20 to 70 m and bedrock velocities of ~5500 m/s associated with Karoo Supergroup strata. Despite the challenges posed by the thick sand cover, lithified calcrete horizons within the Kalahari sediments significantly aided seismic energy propagation. The data were processed using a conventional pre-stack imaging workflow. We tested both Kirchhoff pre-stack time migration (KPreSTM) and Kirchhoff pre-stack depth migration (KPreSDM) and compared the results. Both migration approaches revealed a high degree of similarity in reflector geometries
and structural patterns, suggesting minimal lateral velocity variation across the study area. KPreSTM results were then used in the final seismic interpretation. Pre-stack time migrated sections exhibit nine laterally continuous high-amplitude reflectors between 0.05 and 3.42 km depth, corresponding to major stratigraphic boundaries from the Kalahari Group down to the Ghaap Group. Of particular interest is the moderate-amplitude reflection pair at 1.05–1.35 km depth, interpreted as the Hotazel Formation, the primary host for manganese mineralization. This study demonstrates that, when appropriately designed,
reflection seismic imaging can be a powerful tool for delineating deep mineralized strata beneath thick sedimentary cover in arid environments.

## 1 Introduction

The Paleoproterozoic Kalahari Manganese Field (KMF) (Fig. 1a) of South Africa hosts the world's largest known Manganese (Mn) deposits, accounting for approximately 70 % of the global Mn resources (≈ 5 billion tonnes of ore) (Cairncross and

Beukes, 2013; Beukes et al., 2016; Schnebele, 2023). The initial discovery of these Mn-bearing lithologies was first documented by Rogers (1907). However, due to the remote location and logistical challenges posed by the overlying Kalahari sands and lack of water resources, large-scale exploitation of the deposit was delayed for several decades. Strategic interest in Mn surged during World War II due to global shortages, prompting renewed geological exploration in the region. This led to the opening of the Black Rock Mine by Assmang (Associated Manganese Mines of South Africa) in 1940, followed by the initiation of underground mining operations in 1942 (Cairncross and Beukes, 2013; Beukes et al., 2016).

During the 1950s and 60s, increased geological interest and systematic exploration efforts in the KMF led to the discovery of additional high-grade Mn ore bodies near the town of Hotazel in the Northern Cape province. These discoveries marked a pivotal shift in the regional mining landscape. Exploration drilling and ground magnetic surveys confirmed the economic viability of the deposits, particularly those hosted within the Hotazel Formation (Fig. 1b) of the Postmasburg Group (Beukes et al., 2016). This culminated in the establishment of the Hotazel Mine, strategically located near the sub-outcrop of the formation. Shortly thereafter, the Mamatwan Mine was developed in 1960, becoming South Africa's first major open-pit manganese operation. Unlike the Hotazel deposit, which is known for more structurally controlled and hydrothermally upgraded ore, the Mamatwan deposit is characterized by thick, laterally extensive, and relatively flat-lying stratiform ore bodies. Although the ore grade at Mamatwan is lower (typically ~30–40 wt% Mn), its massive tonnage and ease of extraction make it highly suitable for large-scale open-cast mining. The consistent bedding and shallow depth of burial facilitated the adoption of mechanized mining techniques, making Mamatwan a cornerstone in the growth of the region's manganese industry. This development not only expanded South Africa's share of the global manganese supply but also laid the foundation for long-term infrastructure investments, including roads, railways, and processing facilities that supported subsequent mining operations in the KMF (Cairncross and Beukes, 2013; Beukes et al., 2016).

From the 1980s onward, exploration in the KMF has intensified, characterized by both expansion and diversification. During the 1980s, increased geological reassessment, driven by structural and hydrothermal models, facilitated the discovery of high-grade oxide ore near the N'Chwaning and Gloria mines, expanding the known mineralized footprint of the field (Beukes et al., 2016). In the early 2000s, Assmang further modernized operations by replacing the original N'Chwaning I with successive shafts (N'Chwaning II in 2004 and III in 2006) (Cairncross and Beukes, 2013). Simultaneously, legacy operations received renewed attention. The Smartt Mine, initially developed in 1959, was reopened in 2008 under new ownership, reflecting revived investment interest in previously mined areas (Havenga, 2014). In 2004, the South African government enacted the Mineral and Petroleum Resources Development Act (MPRDA), which transferred mineral rights from private to state ownership. This legislative shift catalyzed a surge in mining and prospecting licenses, expanding the number of active manganese producers in the KMF from two to at least seven by the 2010s (Cairncross and Beukes, 2013).

In November 2023, a high-resolution 2D reflection seismic survey was conducted along five profiles across the Severn farm area (Fig. 1c) over a two-week period. The cost-effective seismic surveys aimed to achieve detailed delineation of the stratigraphic sequence within the KMF, with particular focus on mapping the geometry and continuity of the Hotazel Formation manganese-bearing units. Additionally, the seismic data acquisition sought to identify and characterize key structural features

such as faults, folds, and fractures that potentially influence manganese mineralization and ore body distribution. The seismic technique leveraged a 500 kg mechanical drop hammer as the seismic source and utilized 1C geophones, enabling high spatial resolution imaging despite challenging surface conditions dominated by unconsolidated Kalahari sands and anthropogenic noise.

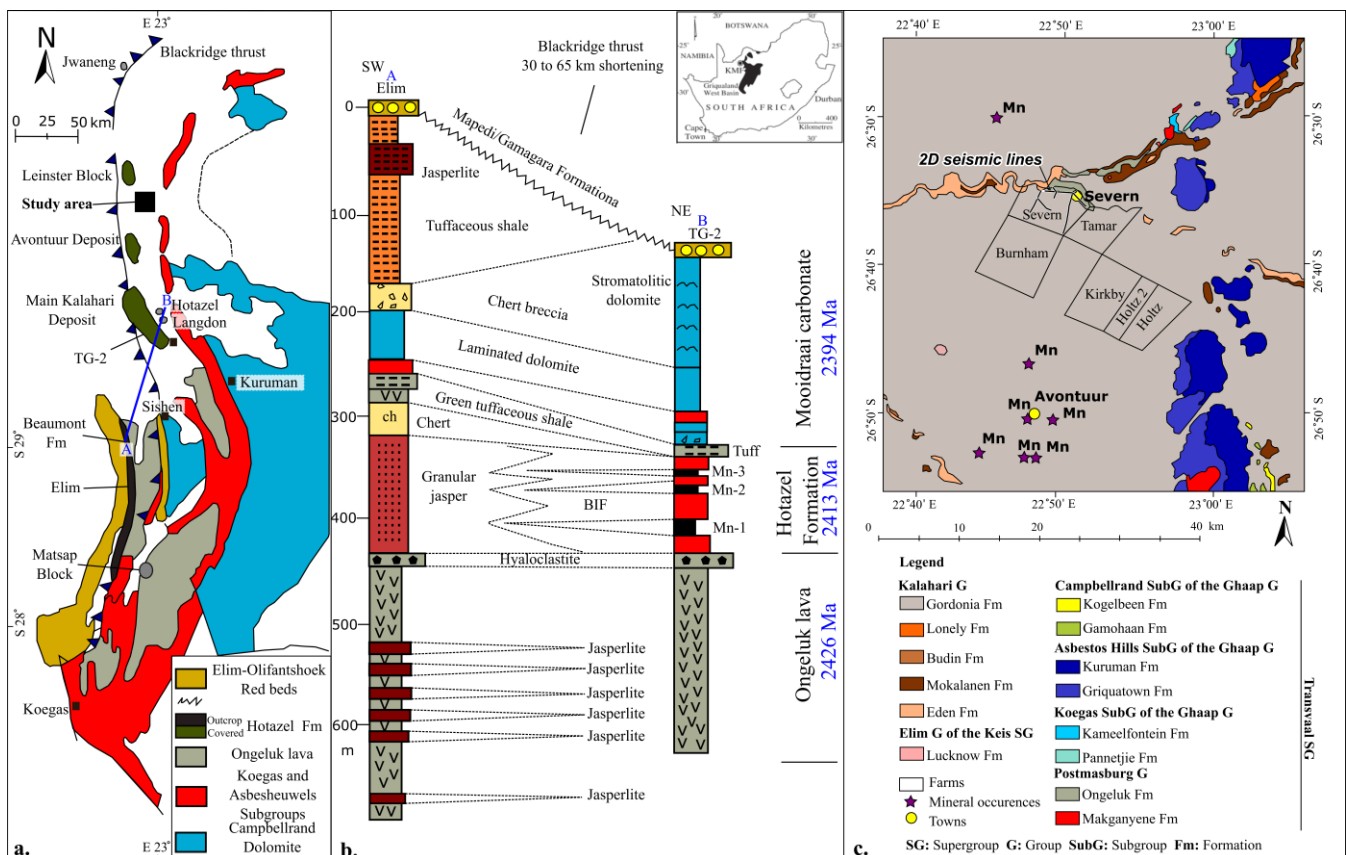

**Figure 1: (a) Regional geological map of the Transvaal Supergroup and lower Keis Supergroup, with the study area indicated by a black square. (b) Lithostratigraphic sections from the Sishen area in the southwest to the Kalahari Manganese Field (KMF) in the northeast, illustrating the stratigraphic relationship of the Ongeluk, Hotazel, and Mooidraai formations (modified after Hongjun et al., 2022; Jogee et al., 2025). (c) Regional geological map of the study area showing the distribution of major lithostratigraphic units and the extensive Kalahari cover (modified after Jogee et al., 2025).**

Complementing the seismic data, an unmanned aerial vehicle (UAV) magnetic survey was undertaken in March and June 2024 (Fig. 2a and 2b) across the study area. This magnetic survey, conducted using a high-sensitivity magnetometer, was designed to detect variations in the earth's magnetic field caused by contrasting lithologies and structural discontinuities. Although magnetic data were acquired to support broader geophysical interpretation, this study focuses primarily on the seismic surveys. The magnetic results are not interpreted here but are intended to be integrated into a future study. Together, these datasets contribute to a comprehensive geophysical framework aimed at improving our understanding of the structural controls on manganese mineralization in the KMF and to inform future exploration efforts in the region.

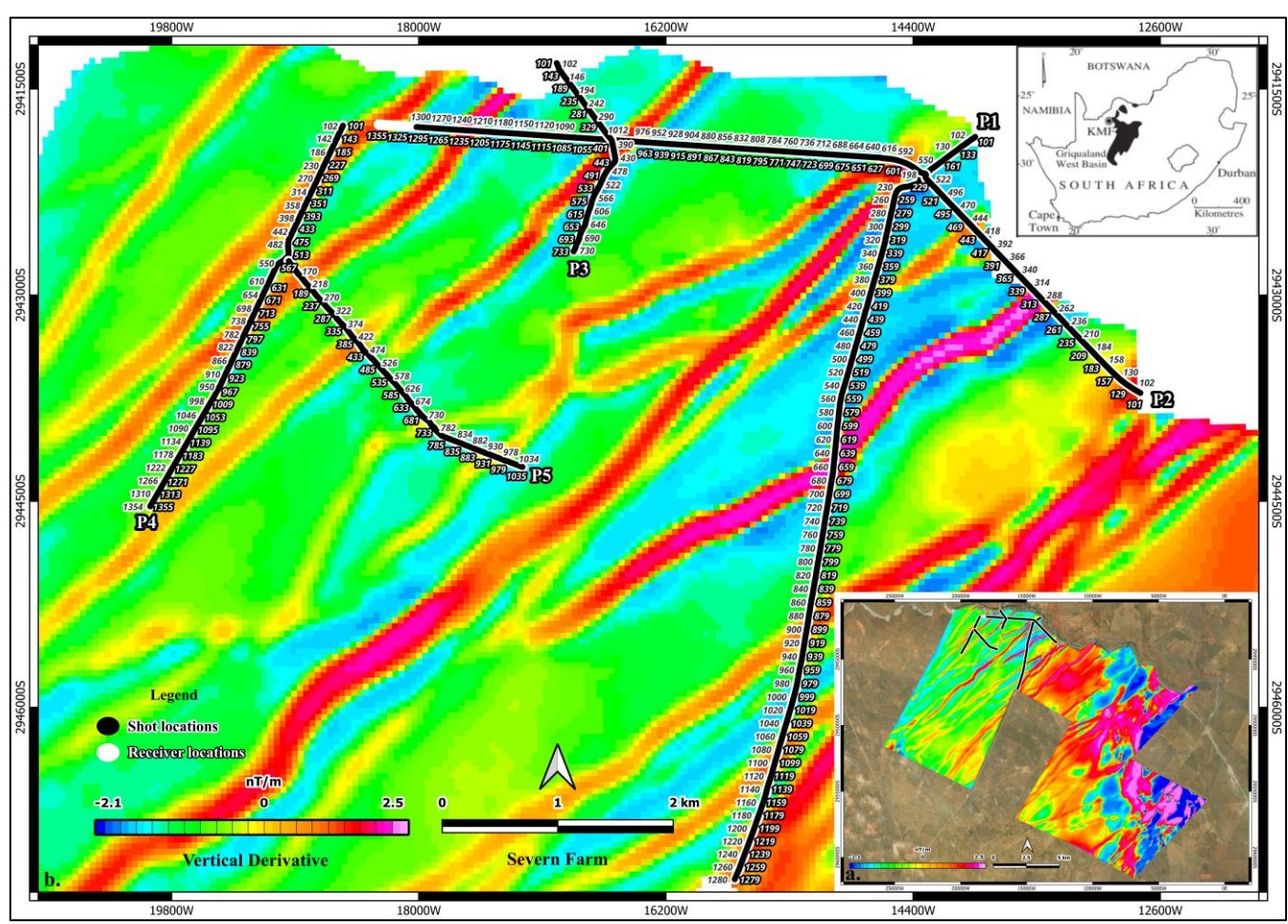

**Figure 2: Seismic acquisition layout in relation to magnetic anomalies across the Kalahari Manganese Field (KMF). (a) First vertical derivative (1VD) of the Total Magnetic Intensity (TMI) map with 2D reflection seismic profiles overlain, illustrating the correlation between subsurface magnetic structures and the seismic survey lines. (b) Zoomed-in view of the Severn farm area showing the detailed layout of the 2D seismic profiles with shot and receiver positions superimposed on the magnetic map. The map highlights localized magnetic anomalies associated with subsurface lithological contrasts relevant to manganese exploration.**

A detailed geological interpretation of these seismic lines has been presented by Jogee et al. (2025). In the present study, we focus on the near-surface characterization using both reflected and refracted wavefield energy to provide an integrated understanding of the shallow subsurface architecture. We further demonstrate how numerical seismic simulations can be effectively utilized to design and optimize data acquisition parameters, assess processing workflows, and guide geological interpretation in complex terranes such as the Griqualand West Basin. In addition, we present and compare results from pre-stack time and depth Kirchhoff migration approaches to evaluate their relative performance in enhancing the imaging of manganese-bearing stratigraphic horizons and associated geological contacts. Finally, we compute the seismic attribute (e.g., cosine of phase) on the seismic sections to enhance the detection of the manganese mineralised zones and compare the results with the traditional seismic amplitude displays.

## 2 Geological Setting

The Kalahari Manganese Field (KMF) is situated in the Northern Cape Province of South Africa, approximately 650 km southwest of Johannesburg (Beukes et al., 2016). These extensive ore bodies are hosted in the Hotazel Formation, a key unit of the Paleoproterozoic Transvaal Supergroup. Within the Northern Cape, the Transvaal Supergroup was deposited in the Griqualand West Basin and is subdivided into the Ghaap and Postmasburg Groups, which together form a thick succession of marine and volcanic strata. These include carbonates, banded iron formations (BIFs), shales, glacial diamictites, basaltic lava flows, and significant manganese ore horizons (Walraven and Martini, 1995; Eriksson et al., 2006). The Transvaal Supergroup is unconformably overlain by the Meso- to Neoproterozoic sedimentary and volcanic sequences of the Keis Supergroup, which themselves are overlain unconformably by the Phanerozoic sediments of the Karoo Supergroup (Beukes et al., 2016; Coetzee et al., 2024).

The Hotazel Formation, part of the Postmasburg Group, is of particular geological and economic importance as it hosts the manganese ore of the KMF. This formation comprises a series of three manganese-rich layers of variable thickness (5 to 10 m, near the study area), each interbedded with BIFs, forming a cyclic sedimentary sequence interpreted as the product of shallow marine depositional environments influenced by redox fluctuations (Beukes et al., 2016; Coetzee et al., 2024; Tsikos et al., 2003). These cycles reflect alternating conditions of iron and manganese precipitation and are critical for understanding the metallogenesis of the region.

Despite its vast mineral potential, the Transvaal Supergroup, including the Hotazel Formation, is largely obscured beneath a widespread cover of younger Cretaceous to Cenozoic sediments that constitute the Kalahari Group. These overlying units are part of the broader Kalahari Basin and comprise unconsolidated to semi-consolidated aeolian sand, pedogenic calcrete, fluvial gravels, and lacustrine clays (Thomas and Shaw, 1991; Haddon and McCarthy, 2005; Beukes et al., 2016). The Kalahari sands, derived through aeolian reworking of older sediments under prevailing arid to semi-arid conditions, form a near-continuous blanket across much of southern Africa and locally reach thicknesses exceeding 100 meters. These sands were transported by ancient river systems and subsequently reworked by wind, resulting in widespread dune fields and sheet sands that dominate the landscape (Lancaster, 2000; Thomas and Shaw, 2002).

In semi-arid regions like the Northern Cape, the upper portions of the Kalahari sedimentary cover have undergone pedogenic lithification to form calcrete, a calcium carbonate-rich indurated layer that results from the accumulation and precipitation of carbonate from percolating meteoric waters (Verboom, 1974; Bond, 1948). These calcrete horizons not only influence surface hydrology and soil development but also significantly affect subsurface geophysical properties. The basal Kalahari sediments, typically consisting of fluvial gravels and silty to clay-rich lacustrine deposits, were emplaced during earlier wetter climatic phases and often act as aquitards, impeding vertical groundwater flow (Pachero, 1976; du Plessis, 1993).

From a geophysical perspective, the nature of the Kalahari cover is particularly relevant. While unconsolidated sands typically scatter and attenuate seismic energy, making it difficult to image deeper stratigraphy, the presence of indurated calcrete layers within the study area has proven advantageous (White et al., 2009). These layers form a more coherent and consolidated near-

surface medium that facilitates the transmission of seismic waves, enhancing the depth of penetration and improving the resolution of the concealed mineralized horizons (Sheriff and Geldart, 1995). This has been crucial in enabling the acquisition of high-resolution seismic reflection data using a small seismic source in otherwise challenging near-surface environments. However, the extensive cover of the Kalahari Group still poses a major obstacle to traditional geological mapping and surface-based exploration methods, as it obscures the outcrop expression of the underlying mineralized Transvaal Supergroup strata (Fig. 1b). As a result, indirect methods such as high-resolution seismic reflection profiling, UAV-based magnetic surveys, and satellite remote sensing are indispensable tools for delineating structural features and stratigraphic boundaries beneath the cover. These methods allow for improved subsurface characterization and targeting of manganese mineralization within the Hotazel Formation.

The manganese deposits of the KMF are confined to the Hotazel Formation of the Transvaal Supergroup, which lies between the mafic Ongeluk lavas and the overlying Mooidraai dolomites. Composed of finely laminated manganese- and iron-rich banded iron formations, the Hotazel Formation hosts dense and competent ore minerals such as braunite, hausmannite, and jacobsite (Beukes et al., 2016; Tsikos and Moore, 1997). Although direct petrophysical measurements were unavailable, the high densities and moderate to high seismic velocities typical of these mineral assemblages are expected to produce strong impedance contrasts with adjacent units. These contrasts enhance reflection seismic responses, especially where mineralized zones are thickened or structurally focused, making the Hotazel Formation a suitable target for seismic imaging in the KMF (Westgate et al., 2020).

In summary, the KMF is not only a world-class manganese province in terms of resource size and ore quality but also presents unique challenges and opportunities for exploration due to its burial beneath a complex, semi-consolidated Kalahari sedimentary cover. Understanding the stratigraphic and geotechnical properties of this overburden is critical for the success of ongoing and future mineral exploration programs in the region.

## 3 Seismic Data

### 3.1 Acquisition

Five 2D reflection seismic profiles were collected in the Severn farm with a combined length of 18,930 m (Fig. 1b) (Jogee et al., 2025). A summary of the seismic data acquisition parameters is shown in Table 1. A 500 kg drop hammer mounted on a compact Bobcat utility vehicle was selected as the seismic source, offering several operational and technical advantages that made it especially suitable for the project area. The small and highly manoeuvrable nature of the Bobcat allowed for efficient movement across the loose, sandy terrain typical of the Kalahari region, significantly improving data acquisition time. In areas with sparse vegetation or minor topographic variations, the Bobcat became bogged down in thick sand; the lightness and compactness of the setup allowed it to be easily extracted using a standard 4×4 vehicle (bakkie), minimizing downtime and maintaining acquisition continuity (Fig. 3). The surface conditions in the study area were not favourable for small 2×4 vehicles, which frequently became stuck in the loose, dry Kalahari sand due to insufficient traction and clearance (Fig. 3a).

**Table 1: Summary of acquisition parameters for the 2D reflection seismic profiles (Mn_P1–Mn_P5) acquired across the Severn Farm area in the Kalahari Manganese Field, South Africa (Jogee et al., 2025).**

**SURVEY PARAMETERS**

| PROFILE | Mn_P1 | Mn_P2 | Mn_P3 | Mn_P4 | Mn_P5 |
|---|---|---|---|---|---|
| Acquisition system | Sercel Lite | Sercel Lite | Sercel Lite | Sercel Lite | Sercel Lite |
| Profile length | 5 890 m | 5 990 m | 1580 m | 3135 m | 2335 m |
| Sampling rate | 2 ms | 2 ms | 2 ms | 2 ms | 2 ms |
| No. of receivers | 590 | 600 | 317 | 628 | 468 |
| Receiver spacing | 10 m | 10 m | 5 m | 5 m | 5 m |
| Geophone | 5 Hz | 5 Hz | 5 Hz | 5 Hz | 5 Hz |
| No. of shots | 590 | 600 | 161 | 314 | 234 |
| Shot spacing | 10 m | 10 m | 10 m | 10 m | 10 m |
| Source type | 500 kg Drop hammer | | | | |
| Geodetic surveying | DGPS | | | | |
| FFID | 4137 - 6524 | 6833 - 9247 | 10212 - 10857 | 10858 - 12113 | 12114 - 13050 |

Data were recorded using 1C, 5 Hz geophones connected to wireless nodes, which enabled efficient burial beneath the loose, dry surface sand to improve geophone-ground coupling and data fidelity (Fig. 3b). The receivers were buried not only to improve coupling in the loose surface sand but also to protect the geophone cables from livestock present on the farm, which were known to chew through exposed wiring. Receiver spacing was 5 m in the central part of the farm (P4 and P5), where the seismic source was continuously pulled due to the presence of thick, loose sand (Fig. 3c), and 10 m along main roads (P1 to P3). Shots were fired every 10 m, with four records stacked at each shot point to enhance signal-to-noise ratio, yielding 1,899 shot points and 4,943,097 traces .

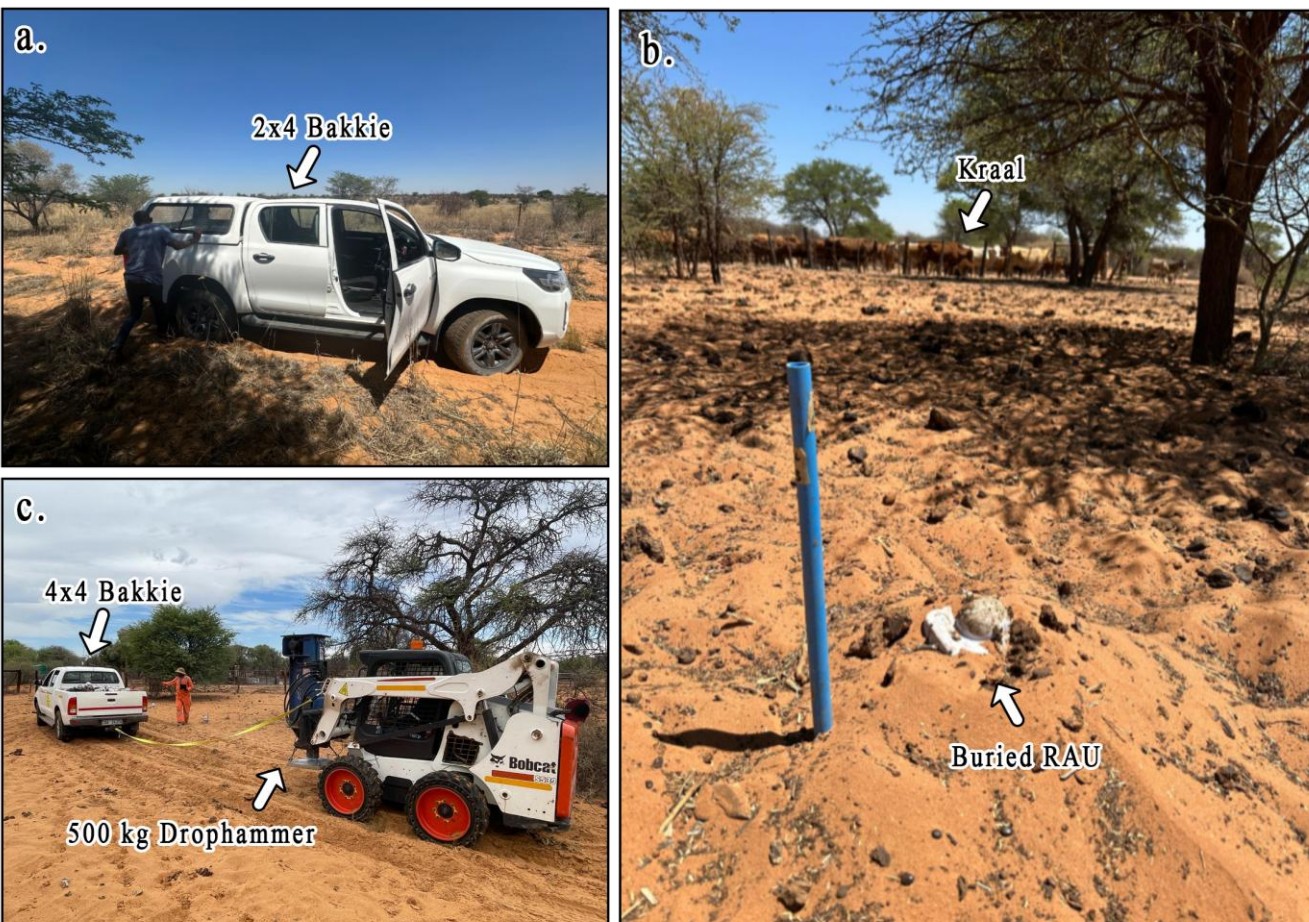

**Figure 3: Field deployment of seismic acquisition equipment under challenging terrain conditions in the Severn farm area. (a) A 2×4 bakkie stuck in soft Kalahari sand, highlighting the need for specialized equipment for mobility. (b) A buried 5 Hz vertical geophone connected to a wireless node (RAU), positioned within the farm to prevent livestock interference. (c) A 500 kg drop hammer mounted on a compact Bobcat and towed by a 4×4 bakkie for improved manoeuvrability across sandy terrain.**

An example shot gather from profile P5 is presented in Fig. 4a, showing well-defined first arrivals (green arrows in Fig. 4a), while the corresponding frequency spectrum (Fig. 4b) reveals dominant low-frequency energy in the range of 8 to 14 Hz. The background noise (shaded blue in Fig. 4a), ground roll (shaded orange in Fig. 4a), and guided wave (shaded yellow in Fig. 4a) energy are highlighted in the shot gather, illustrating the different types of waveforms captured during acquisition.

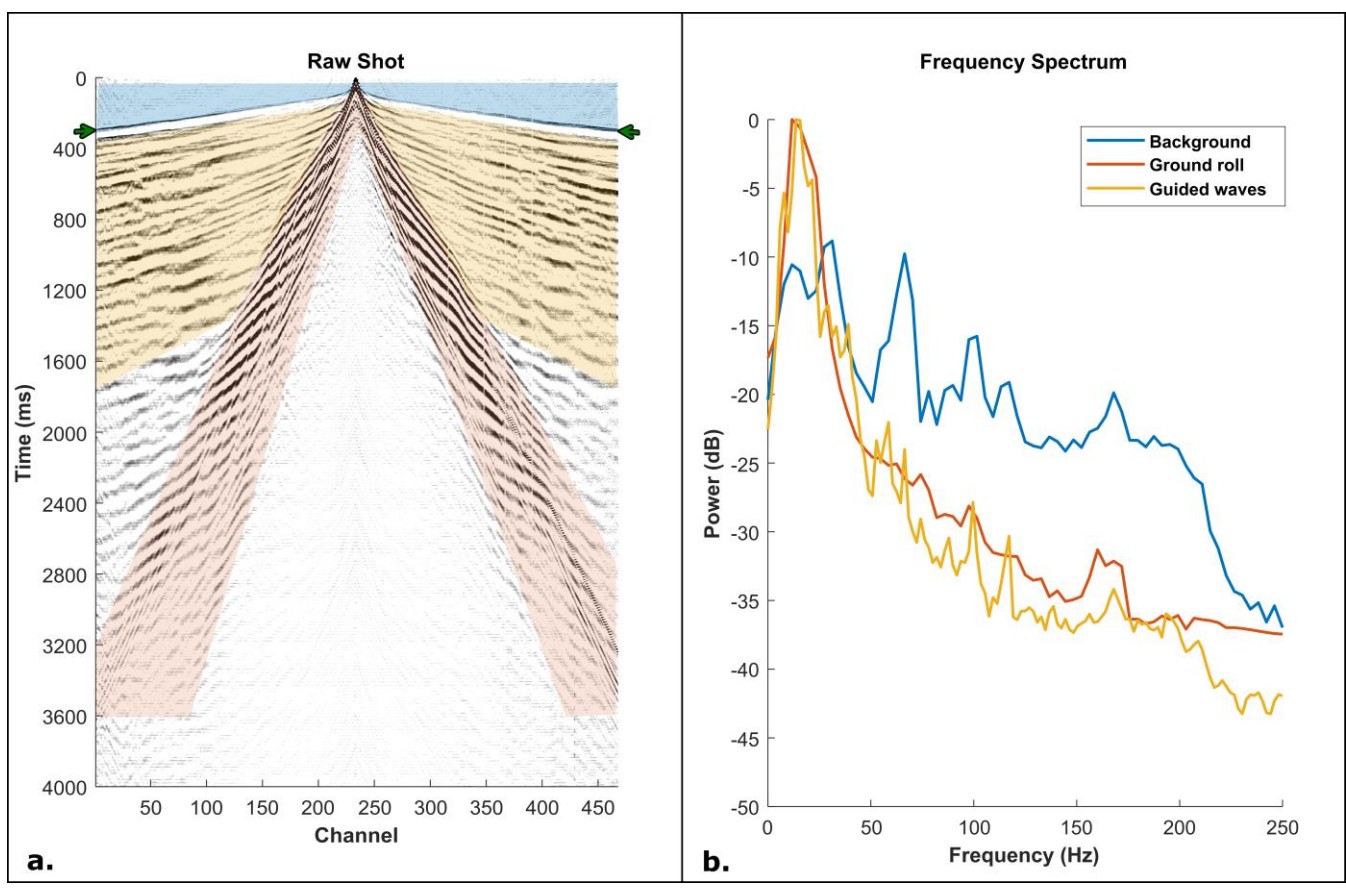

**Figure 4: a) Example shot gather from profile 5 illustrating key components of the recorded wavefield. Background noise is shaded in blue, guided waves in yellow, and ground roll in orange, as well as clear first breaks shown by the green arrows. (b) Corresponding frequency spectrum graph showing dominant low-frequency energy concentrated between 8–14 Hz, characteristic of surface wave propagation in the unconsolidated near-surface Kalahari sediments.**

## 3.2 Processing

A variable presence of low-frequency energy (8–14 Hz) is observed across the five seismic profiles, with profiles 1 to 3 exhibiting a relatively weaker presence of this signal, while profiles 4 and 5 are almost entirely dominated by it. Fig. 5 presents examples of raw and processed shot gathers from each profile with trace amplitudes balanced for visualisation, demonstrating this variation in low-frequency content. The example shot gathers correspond to the intersection points of the profiles, allowing for direct comparison between profiles. In profiles 1 (Fig. 5a) and 2 (Fig. 5b), low signal penetration and weak first breaks at far offsets are observed, suggesting unfavourable near-surface conditions at the intersection of these profiles. Profile 3 is affected by strong 50 Hz electrical noise originating from overhead power lines, which is shaded blue in the Fig. 5c. This interference obscures a significant portion of the recorded signal. For profiles 4 (Fig. 5d) and 5 (Fig. 5e), the reflected wavefield is masked by dominant low-frequency energy, possibly linked to surface wave propagation through unconsolidated or heterogeneously consolidated sand and calcrete layers. This strong low-frequency content significantly masks the deeper

reflectors, complicating processing and interpretation. The ground roll energy is highlighted by red arrows in the shot gathers. The associated frequency spectra for each profile are shown in Fig. 5k to 5o. These observations underscore the spatial heterogeneity in near-surface conditions and highlight the importance of adaptive processing strategies to suppress noise and enhance signal quality.

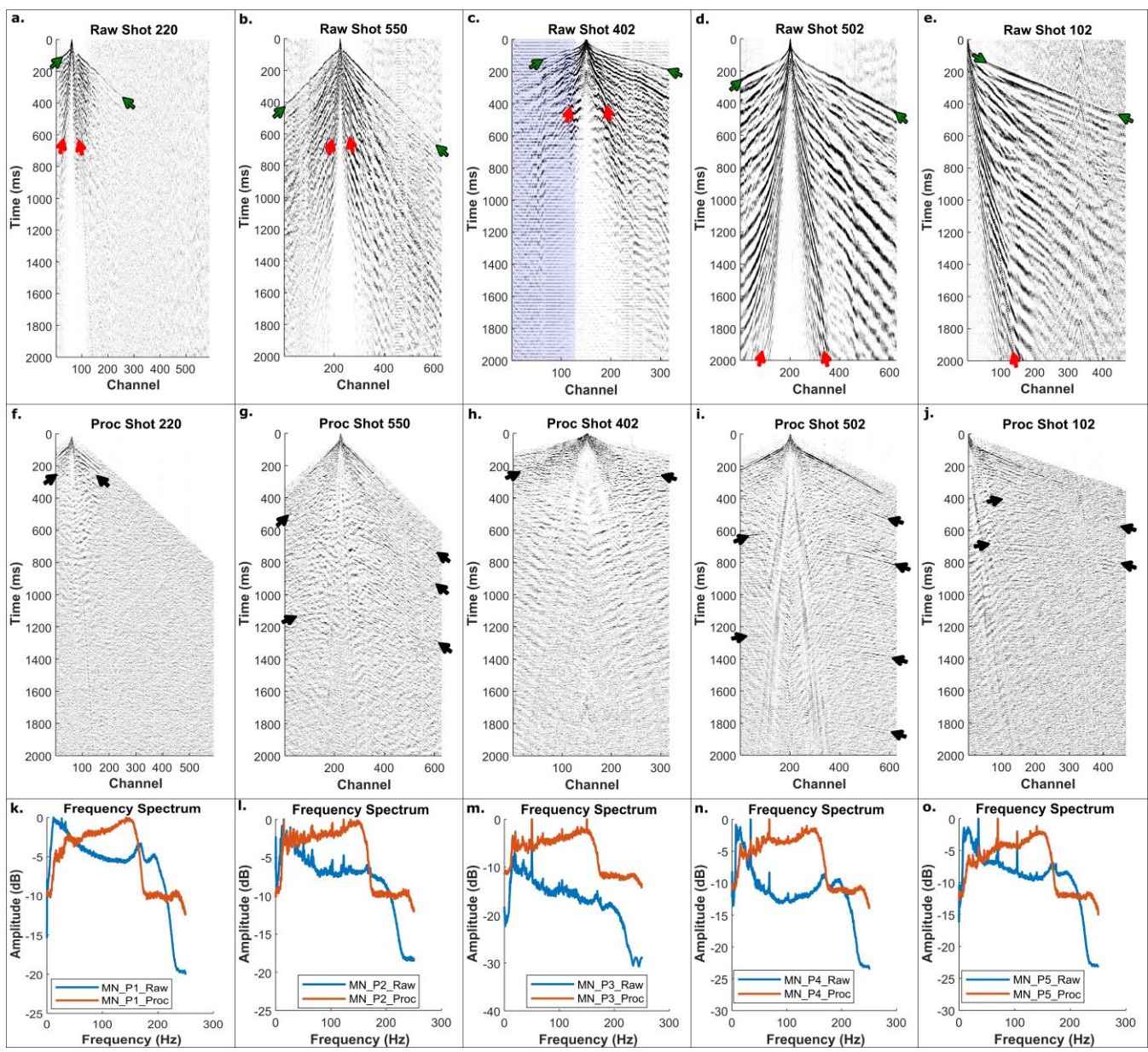


**Figure 5: Example raw (a to e) and processed (f to j) shot gathers from seismic profiles 1 to 5, respectively. Each panel highlights characteristic wavefield components: clear first arrivals (green arrows), reflected wavefield energy (black arrows), ground roll (red**

arrows), and 50 Hz electrical noise (shaded blue). The corresponding frequency spectrum graphs are shown in panels (f–j), illustrating variations in dominant frequency content across the survey lines.

The reflection seismic data were processed using a conventional workflow, incorporating key stages such as first-break picking, refraction statics correction, noise attenuation, deconvolution, velocity analysis, pre-stack migration, stacking, and post-stack enhancement. A summary of the seismic data processing workflow is shown in Table 2. This systematic approach ensured improved signal-to-noise ratio, accurate positioning of reflection events, and enhanced resolution of subsurface features critical for geological interpretation.

In summary, the seismic data processing workflow commenced with geometry definition, which involved establishing a crooked-line geometry and common depth point (CDP) binning. The CDP bin size along the profile was defined as half of the receiver spacing, while the bin spacing was set to 40% greater than half the bin size. Following geometry setup, automated first-break picking followed by manual refinement, with calculated refraction statics yielding RMS errors up to 5 ms and data corrected to a final datum elevation per profile. Ground roll noise and guided-wave noise, dominant in the shot domain, were

effectively attenuated using a radial trace transform (RTT) filter. The seismic data were transformed from the spatial-temporal $(x - t)$ domain into the radial trace $(\tau - v_r)$ domain (Claerbout, 1975), where guided waves were characterized using a low-pass filter based on their low-frequency attributes. The identified guided-wave components were then transformed back into the $(x - t)$ domain and subtracted from the data to suppress their influence while preserving true reflections. To further attenuate residual guided-wave energy, FK (frequency–wavenumber) filtering was subsequently applied.

Linear noise and airwaves were attenuated using horizontal median and airwave filters, respectively. Reflection quality was enhanced through Wiener deconvolution and bandpass filtering (10–150 Hz), followed by multiple rounds of velocity analysis and surface-consistent residual statics. Isotropic Kirchhoff pre-stack time and depth migration was applied to accurately position reflections, and post-stack processing included spherical divergence correction, Tau-P coherency filtering, and spectral balancing to further improve reflection continuity and resolution for geological interpretation (Jogee et al., 2025).

Post-migration, we also computed the complex-trace attribute to enhance the quality of the seismic interpretation (Manzi et al., 2013). Complex-trace seismic attributes are derived from the seismic data (along seismic traces) through mathematical manipulation of the wave components, such as the phase, frequency, and amplitude. Amplitude-independent seismic attributes are generally effective at enhancing the continuity and extent of the weak seismic reflections. In this study, the cosine of the phase attribute (Manzi et al., 2013) was computed along the seismic traces to map the continuity of the seismic reflections with

particular focus on enhancing the detection of the mineralized zone and associated discontinuities such as faults.

**Table 2: Summary of the processing steps applied to the reflection seismic datasets for profiles P1, P2, P3, P4, and P5 (Jogee et al., 2025).**

| STEP | PROCESS | PARAMETERS |
|---|---|---|
| 1 | Data input | Read SEGD data, convert to HDF5 |
| 2 | Stack shots | 4 repeated shot records, diversity stacked |

| 3 | Geometry setup | 5 m (P1-P2), and 2.5 m (P1-P5) CMP spacing, crooked line binning |
|---|---|---|
| 4 | Trace editing | Remove noisy, dead, and unbinned traces |
| 5 | First break picking | All available offsets |
| 6 | Refraction statics | P1: Datum: 1030 m, replacement velocity: 1000 m/s |
| | | P2: Datum: 1030 m, replacement velocity: 1000 m/s |
| | | P3: Datum: 1015 m, replacement velocity: 1000 m/s |
| | | P4: Datum: 1025 m, replacement velocity: 1000 m/s |
| | | P5: Datum: 1025 m, replacement velocity: 1000 m/s |
| 7 | Radial trace filter | Vrange: -6500:6500 m/s and Lowpass filter: 8 – 18 Hz |
| 8 | Horizontal median filter | Steering velocity: 3000 m/s |
| 8 | Airwave filter | Velocity: 350 m/s |
| 9 | Wiener deconvolution | Gap: 2 ms and length: 90 ms |
| 10 | Bandpass filter | 4 – 10 – 150 – 200 Hz |
| 11 | Residual statics | Surface consistent (looped) |
| 12 | Migration | Isotropic Kirchhoff pre-stack time and depth migration |
| 13 | NMO and stacking | Interval velocity model and Normalise (Squar Root normalisation) and Smooth stack |
| 14 | Post-stack processing (Spherical divergence, Semblance smoothing, and Spectral Weighting) | (Grid controlled velocity interpolation, Tau-P coherency filter, and 5 – 150 Hz frequencies scaled by factors from 0.25 – 3) |
| 15 | Time-to-depth conversion | Smoothed migration velocities |

## 3.3 Refraction Tomography

To investigate the thickness and lateral variability of the Kalahari Group sediments across the Severn project area, refraction
tomography was applied using first-arrival travel times extracted from the surface seismic dataset. This method provided robust
near-surface velocity models that offer critical insights into both the subsurface structure and material properties that influence
seismic wave propagation. The analysis was conducted using a nonlinear travel time inversion algorithm developed by Zhang
and Toksöz (1998), which employs ray tracing to iteratively match synthetic and observed traveltimes, refining the subsurface
velocity model. The initial model consisted of three layers, constructed by fitting the observed gradients in the picked P-wave
first arrivals, and was refined through successive iterations to minimize root-mean-square (RMS) errors between observed and
calculated travel times.

Fig. 6 shows the tomographic P-wave velocity models for seismic profiles 1 to 5, along with their associated ray paths. The
ray coverage illustrates the distribution and density of seismic rays used in the inversion process, highlighting areas of good

resolution within the velocity models. These models clearly delineate the low-velocity near-surface layers corresponding to

the Kalahari Group sediments and the high-velocity Karoo Supergroup bedrock interface, providing key constraints on sediment thickness and lateral variability across the study area.

The velocity models generated from this tomographic inversion revealed significant spatial variations in the thickness and seismic velocities of the Kalahari Group cover sediments (black arrows and dotted lines in Fig. 6). Profiles 1, 2, and 3 were acquired along compacted gravel roads, where the Kalahari Group cover is comparatively thin, ranging from approximately

20 to 40 meters in total thickness. Specifically, profile 1 has an estimated total Kalahari cover thickness of around 40 meters (Fig. 6a), of which approximately 20 meters comprises aeolian sand (Fig. 6a). Profiles 2 and 3 each show a reduced total Kalahari thickness of approximately 20 meters, with 10 meters of aeolian cover (Fig. 6b, c).

These thinner sediment packages and the presence of compacted road surfaces facilitated better coupling of the geophones and higher-frequency seismic wave penetration, resulting in improved resolution of reflected wavefields in the seismic data.

Additionally, profile 2 crosses an outcrop of Ongeluk Formation basaltic lavas, dense, high-velocity units. Although basalts are generally known to attenuate and scatter seismic energy, especially when overlying target horizons, the shallow and laterally restricted nature of this outcrop likely limited such effects. Its presence may have improved geophone coupling and supported the transmission of higher-frequency energy into the subsurface, thereby enhancing the resolution of near-surface reflections in this profile (Sun et al., 2010).

In contrast, profiles 4 and 5 were situated within the interior of the farm, away from road infrastructure, where Kalahari Group sediments are significantly thicker. Profile 4 reaches a maximum total Kalahari Group thickness of approximately 60 meters, with around 20 meters attributed to aeolian sands (Fig. 6d). Profile 5 shows the greatest thickness of Kalahari Group sediments, reaching approximately 70 meters, also with a 20-meter-thick aeolian cap (Fig. 6e). The thicker, low-velocity sand and clay-rich sediments (ranging from 600 to 4000 m/s) in these areas act as seismic waveguides, trapping surface wave energy and

giving rise to pronounced ground roll and guided waves. These wave types dominate the recorded data at low frequencies (8–14 Hz), which obscure deeper reflections in the raw shot gathers. This effect is most prominent in the low frequency dominated profiles (P4 and P5), in contrast to the clearer, more resolved reflections observed in profiles 1 and 2.

Beneath this sedimentary cover, the underlying Karoo Supergroup strata constitute the seismic bedrock, displaying P-wave velocities of approximately 5500 m/s. This significant contrast in seismic velocity between the unconsolidated to semi-

consolidated Kalahari cover and the consolidated Karoo bedrock creates a well-defined refractor in the tomography models. The identification of this high-velocity horizon is critical for accurate static corrections, velocity model building, and interpretation of deeper seismic reflections. These tomographic models not only help constrain the structural and stratigraphic configuration of the near surface but also highlight the influence of variable sediment cover on seismic imaging quality, especially in mineral exploration settings where buried targets are masked by thick surficial deposits.

The tomographic results reveal that the near-surface Kalahari Group sediments generally exhibit low P-wave velocities, ranging from approximately 600 m/s to 4000 m/s. The shallowest portions of the model, where velocities fall below 1000 m/s, are interpreted as unconsolidated to semi-consolidated alluvial and aeolian sands and gravels of the Kalahari Group (Thomas

and Shaw, 1991; Haddon and McCarthy, 2005; Beukes et al., 2016). These low-velocity units are laterally variable, with their thickness and depth extent decreasing notably from the interior farm profiles (P4 and P5) toward the road-based profiles (P1

and P2), where the aeolian cover is considerably thinner.

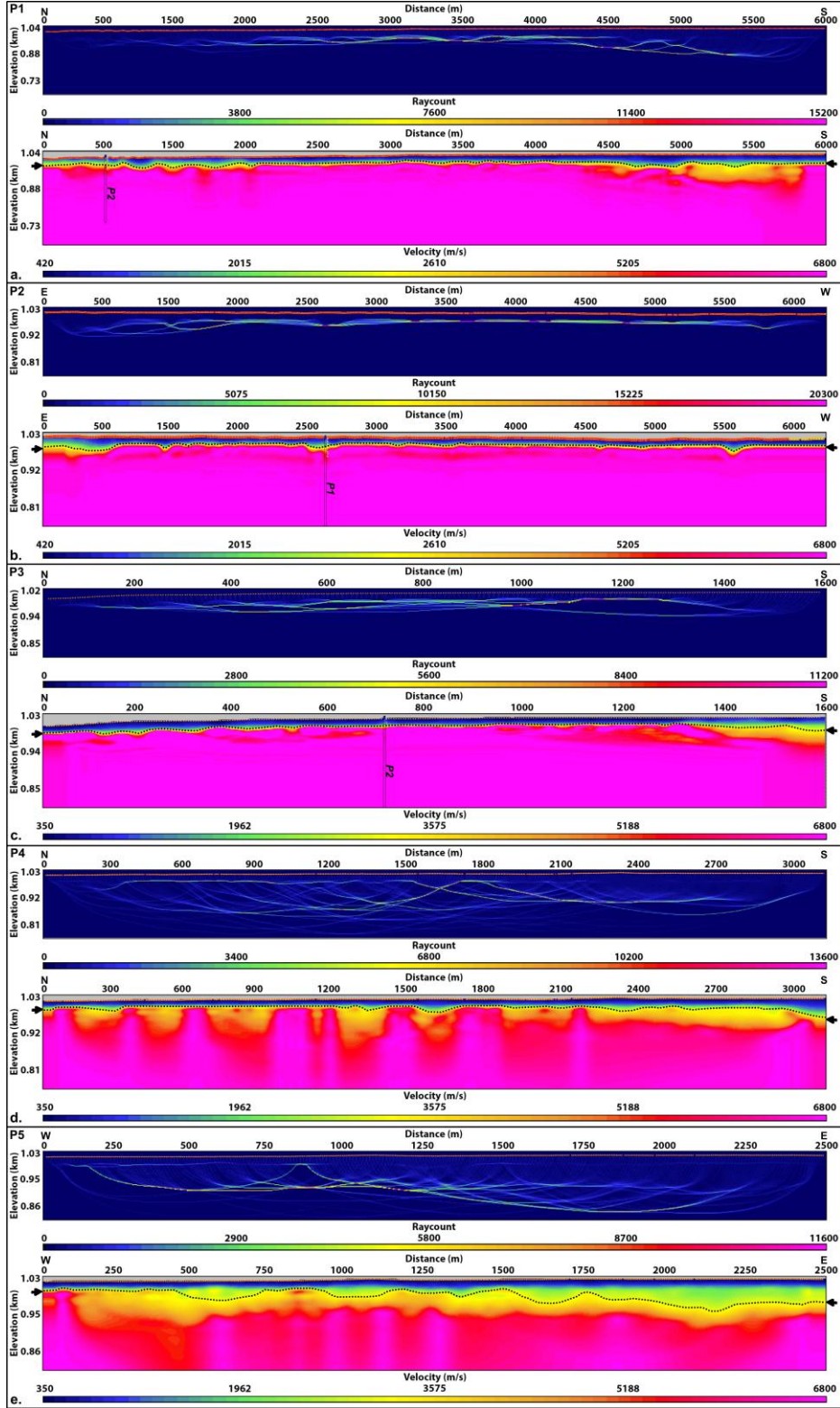


**Figure 6: Refraction tomographic P-wave velocity models with associated ray coverage for seismic profiles 1 to 5 (a–e, respectively). The models display near-surface velocity variations, with velocities ranging from 600 to 5500 m/s. The base of the Kalahari Group sediments is interpreted and indicated by black arrows and doted lines.**

## 4 Results and Discussion

Fig. 7a–7d and 7e–7h present a comparative assessment of the Kirchhoff pre-stack time (KPreSTM) and depth (KPreSDM) migration results for profiles 1, 2, 4, and 5, respectively. To enable a direct and meaningful comparison, both migration approaches utilized the same velocity models derived from the respective profiles, which are overlain on the sections in Fig.

7. At shallow depths (0–800 m), particularly above the strong continuous reflector indicated by the blue arrow, the KPreSTM sections exhibit clearer reflection continuity. At greater depths, both migration methods produce broadly comparable reflection patterns. However, the KPreSDM results more effectively delineate steeply dipping and cross-cutting geological structures, whereas the KPreSTM images display enhanced continuity along near-horizontal horizons.

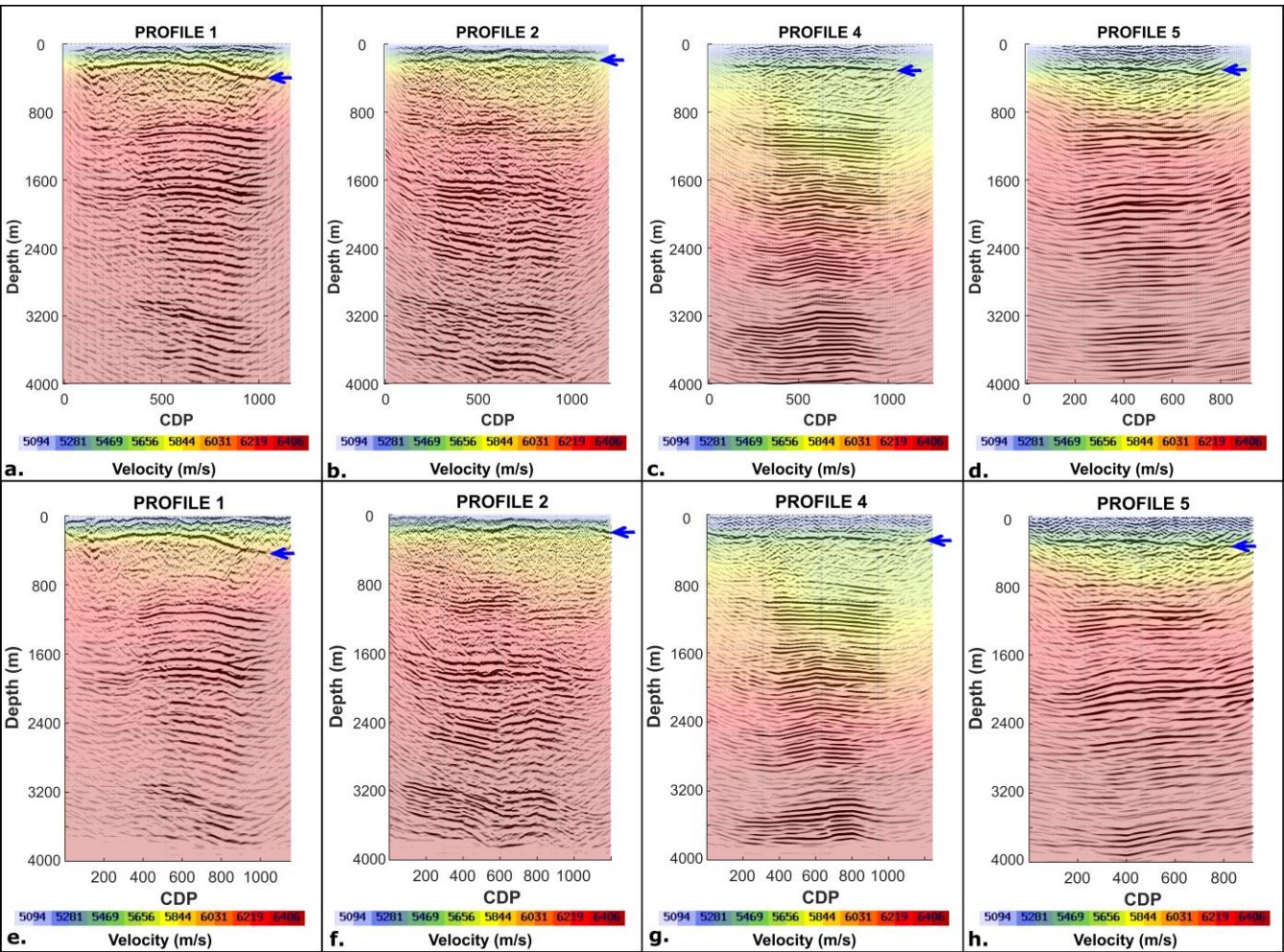

 **Figure 7: Comparison of Kirchhoff pre-stack time migration (KPreSTM; a–d) and Kirchhoff pre-stack depth migration (KPreSDM; e–h) results for seismic profiles 1, 2, 4, and 5. The migration velocity models are overlain on each section.**

As highlighted by Etgen and Kumar (2012), there is no inherent theoretical difference between PreSTM and PreSDM in their ability to produce high-resolution or amplitude-preserving images. Any discrepancies between the two approaches primarily arise from differences in velocity model construction, the treatment of lateral velocity variations, and the numerical approximations used in their respective implementations. In this study, the comparable imaging results between the KPreSTM and KPreSDM sections likely reflect the absence of significant lateral velocity variations within the survey area and the overall simplicity of the velocity field. The dominance of near-horizontal reflections and the presence of laterally continuous stratigraphy further favour the performance of PreSTM, as under such conditions, depth migration offers limited additional improvement in reflector positioning or structural delineation. Consequently, the strong correspondence between the PreSTM and PreSDM results suggests that the applied time-domain velocity models are sufficiently representative of the true subsurface geometry.

The processed reflection seismic profiles (Lines 1, 2, 4, and 5) reveal nine prominent, laterally continuous, high-amplitude reflections interpreted at depths of approximately 50 m, 300 m, 700 m, 1100 m, 1350 m, 1700 m, 2240 m, 2810 m, and 3420 m (Fig. 8). These reflectors, numbered 1 to 9 from bottom to top, correspond to major lithostratigraphic boundaries beneath the Kalahari Manganese Field (KMF), spanning the crystalline basement through the Transvaal, Keis, and Karoo Supergroups up to the near-surface Kalahari Group. Each reflector marks a transition between units with contrasting acoustic impedances, driven by changes in lithology, density ($\rho$), and seismic P-wave velocity ($V_p$):

1.  Ventersdorp Supergroup – Campbellrand and Schmidtsdrif contact: Lavas ($\rho \approx 2.9$ g/cm³; $V_p \approx 6.4$ km/s) – Dolomitic carbonates ($\rho \approx 2.8$ g/cm³; $V_p \approx 5.8$–6.0 km/s) overlying deeper clastic and carbonate sediments ($\rho \approx 2.5$ g/cm³; $V_p \approx 4.8$–5.2 km/s) of the Transvaal Supergroup (Tinker et al., 2002; Beukes et al., 2016).

2.  Campbellrand and Schmidtsdrif – Asbestos Hills and Koegas contact: Iron-rich BIFs ($\rho \approx 3.3$–3.6 g/cm³; $V_p \approx 5.8$–6.0 km/s) transitioning into shale-dominated units ($\rho \approx 2.4$–2.6 g/cm³; $V_p \approx 3.5$–4.5 km/s) (Tsikos and Moore, 1997; Beukes et al., 2016; Westgate et al., 2020).

3.  Ghaap – Postmasburg contact: Carbonates to volcanogenic sediments, with an estimated velocity increase from ~5.0 to 6.0 km/s and density from ~2.6 to 2.9 g/cm³ (Beukes et al., 2016; Tinker et al., 2002).

4.  Makganyene – Ongeluk contact: Diamictites ($\rho \approx 2.3$–2.5 g/cm³; $V_p \approx 3.0$–4.0 km/s) overlain by mafic lavas ($\rho \approx 2.9$–3.0 g/cm³; $V_p \approx 5.5$–6.0 km/s) (Reimold et al., 2002; Tinker et al., 2002; Beukes et al., 2016).

5.  Ongeluk – Hotazel contact: Mafic lavas overlying manganese-rich BIFs ($\rho \approx 3.4$–3.6 g/cm³; $V_p \approx 6.2$–6.5 km/s), producing a sharp impedance contrast (Beukes et al., 2016; Tsikos and Moore, 1997).

6.  Hotazel – Mooidraai contact: Dense BIFs beneath stromatolitic dolomites ($\rho \approx 2.7$–2.9 g/cm³; $V_p \approx 5.2$–5.8 km/s) (Beukes et al., 2016; Van Niekerk and Beukes, 2019).

7. Transvaal – Keis Supergroup (pre-Gamagara unconformity): Transition from older carbonate-volcanic strata ($\rho \approx$ 2.8–2.9 g/cm³; Vp $\approx$ 5.5–6.0 km/s) to metasedimentary rocks of the Keis Supergroup ($\rho \approx$ 2.6–2.8 g/cm³; Vp $\approx$ 4.5–5.5 km/s) (Van Niekerk and Beukes, 2019; Westgate et al., 2020).

8. Keis – Karoo Supergroup contact: Folded metasediments ($\rho \approx$ 2.6–2.8 g/cm³; Vp $\approx$ 4.5–5.0 km/s) overlain by Karoo siliciclastic sediments ($\rho \approx$ 2.4–2.6 g/cm³; Vp $\approx$ 3.5–4.5 km/s) (Tinker et al., 2002).

9. Karoo – Kalahari Group contact: A strong near-surface contrast between consolidated Karoo bedrock ($\rho \approx$ 2.5–2.6 g/cm³; Vp $\approx$ 4.0–5.0 km/s) and unconsolidated Kalahari cover ($\rho \approx$ 1.8–2.2 g/cm³; Vp $\approx$ 0.5–2.0 km/s) (Reimold et al., 2002; Tinker et al., 2002).

These velocity and density contrasts produce significant differences in acoustic impedance ($Z = \rho \times Vp$), which account for the strength and continuity of the observed reflections across the seismic profiles. The values used to estimate these contrasts are derived from a combination of borehole sonic log data (e.g., KHK-1 and NEV-1; Reimold et al., 2002; Westgate et al., 2020), tomographic velocity models from this study, and published physical property parameters from prior geological and mineralogical investigations in the Transvaal and Griqualand West basins (Beukes et al., 2016; Tsikos and Moore, 1997;
Tinker et al., 2002). Further support comes from regional seismic studies, including recent work by Westgate et al. (2020, 2021), which documented similar lithological transitions and reflection patterns using legacy 2D seismic data.

The manganese-rich Hotazel Formation, composed predominantly of BIFs and minerals such as braunite, hausmannite, and jacobsite is expected to exhibit bulk densities in the range of 3.4–3.6 g/cm³ and P-wave velocities between 6.2 and 6.5 km/s (Beukes et al., 2016; Tsikos and Moore, 1997). This contrasts with the overlying Mooidraai dolomites and underlying Ongeluk
lavas, which show lower densities (2.7–2.9 g/cm³) and velocities (5.2–5.8 km/s), creating a moderate but distinct acoustic impedance contrast. Similarly, deeper reflections associated with volcanic-carbonate interfaces and basement transitions exhibit stronger contrasts, with velocities exceeding 6.5 km/s and densities >3.0 g/cm³ in the crystalline units (Reimold et al., 2002; Westgate et al., 2020).

These nine reflections thus establish a coherent and stratigraphically significant seismic framework, with near-horizontal dips
(<5°) suggesting limited post-depositional deformation. Notably, the reflectors between ~1100 and 1350 m (contact 6 and 5) depth are interpreted as the top and base of the Hotazel Formation, the primary manganese-hosting unit of the KMF. The moderate reflection amplitudes observed at these depths are consistent with the estimated impedance contrasts between the Hotazel BIFs and their surrounding dolomitic units. This seismic expression is comparable to that seen in reprocessed legacy seismic sections from the western Kaapvaal Craton margin, where stratigraphic BIF contacts and mineralized zones were
imaged under thick cover rocks (Westgate et al., 2020, 2021).

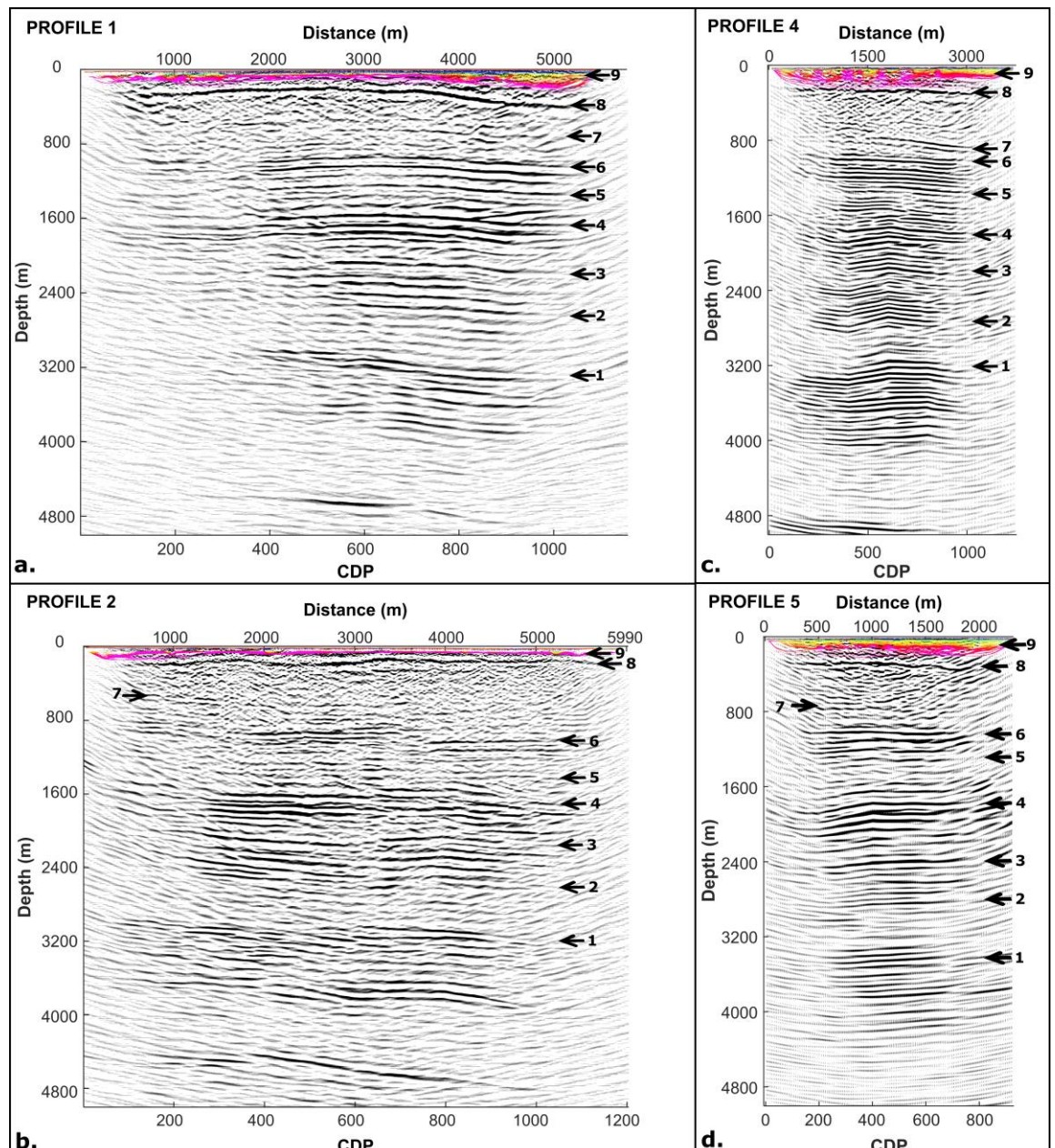

**Figure 8: The Kirchhoff pre-stack time-migrated (PSTM) reflection seismic sections for profiles 1 (a), 2 (b), 4 (c), and 5 (d), with key interpreted reflectors annotated using black arrows. Overlain on each section is the cropped refraction tomographic P-wave velocity model, constrained to the portion of the profile with reliable ray coverage. The seismic sections reveal nine prominent, laterally continuous, high-amplitude reflections, interpreted as major lithostratigraphic contacts across the Kalahari Manganese Field (KMF). These are numbered 1 to 9 from bottom to top and correspond to: (1) the Ventersdorp Supergroup – Campbellrand and Schmidtsdrif Subgroup contact; (2) Campbellrand and Schmidtsdrif – Asbestos Hills and Koegas Subgroup contact; (3) the boundary between the Ghaap and Postmasburg Groups; (4) the Makganyene–Ongeluk Formation contact; (5) the Ongeluk–Hotazel Formation contact; (6) the Hotazel–Mooidraai Formation contact; (7) the pre-Gamagara unconformity between the Transvaal and Keis Supergroups; (8) the Keis–Karoo Supergroup contact; and (9) the surface unconformity between the Karoo Supergroup and the overlying Kalahari Group. These reflectors highlight the internal basin architecture and stratigraphic continuity, with particular**

**emphasis on the Hotazel Formation interval (reflections 5 and 6), which hosts the primary manganese mineralization targeted in this study.**

Fig. 9 shows a one-dimensional (1D) synthetic seismogram generated using average physical properties derived from the interpreted geological contacts on the seismic profiles. The synthetic seismogram was created using a temporal sampling rate of 1 ms and a Ricker wavelet with a dominant frequency of 60 Hz, with no applied attenuation effects. This modelling approach provided a means to validate key reflection events, correlate major stratigraphic boundaries, and enhance confidence in the interpretation of subsurface structures.

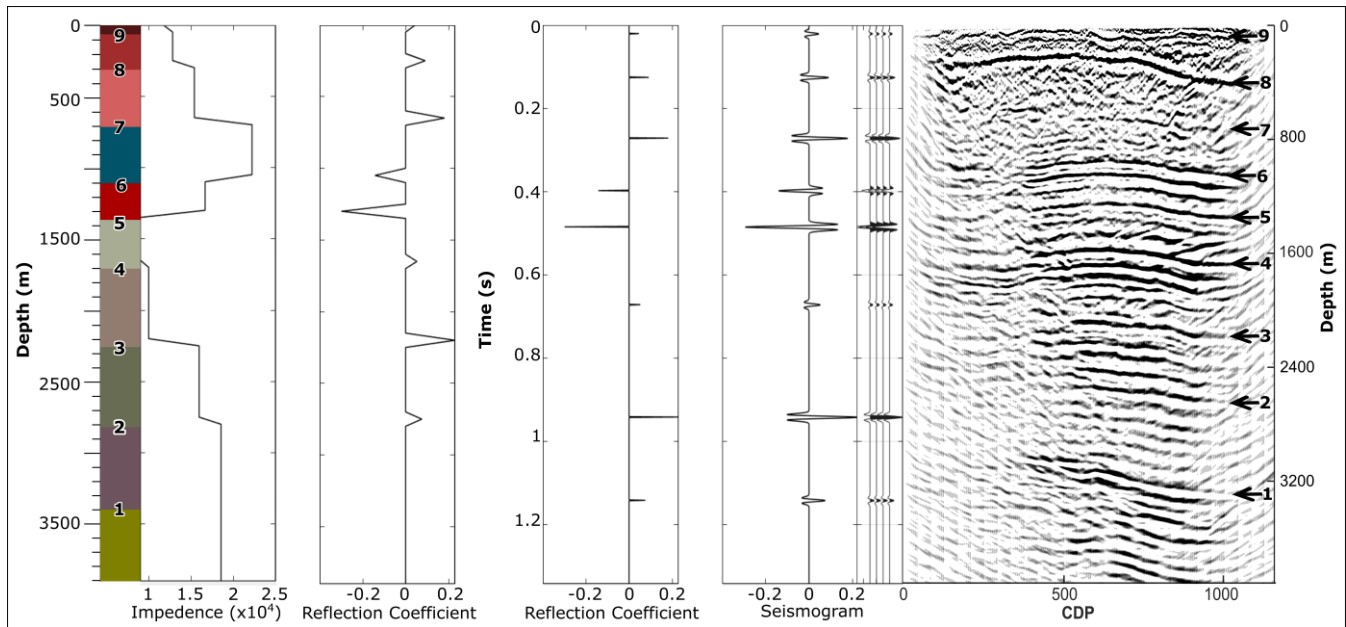

**Figure 9: One-dimensional (1D) synthetic seismogram generated using average physical properties derived from interpreted geological contacts on the seismic profiles, as well as profile 1 showing key reflection contacts numbered 1 to 9 from bottom to top and correspond to: (1) the Ventersdorp Supergroup – Campbellrand and Schmidtsdrif Subgroup contact; (2) Campbellrand and Schmidtsdrif – Asbestos Hills and Koegas Subgroup contact; (3) the boundary between the Ghaap and Postmasburg Groups; (4) the Makganyene–Ongeluk Formation contact; (5) the Ongeluk–Hotazel Formation contact; (6) the Hotazel–Mooidraai Formation contact; (7) the pre-Gamagara unconformity between the Transvaal and Keis Supergroups; (8) the Keis–Karoo Supergroup contact; and (9) the surface unconformity between the Karoo Supergroup and the overlying Kalahari Group.**

Fig. 10 presents the stacked reflection seismic sections for profiles 1 2, 4, and 5, overlaid with their corresponding near-surface P-wave velocity models derived from refraction tomography. For clarity, the tomographic models are truncated to include only the sections with sufficient ray coverage. These near-surface velocity models provide critical insights into the lithological and structural variability of the uppermost Kalahari Group sediments across the Severn survey area.

The seismic profiles also resolve a key stratigraphic boundary, the contact between the Karoo Supergroup and the underlying Keis Supergroup (Reflector 8). This contact is observed to occur at shallower depths in profiles 1 and 2 (Fig. 10a, b), which correspond to the more elevated and sand-deficient regions of the survey area. In contrast, this contact is much deeper in profiles 4 and 5 (Fig. 10c, d), where the overlying Kalahari Group sediments are thicker. These observations are consistent

with the refraction tomography-derived thickness estimates and support the interpretation that the aeolian cover becomes progressively thicker toward the central parts of the farm.

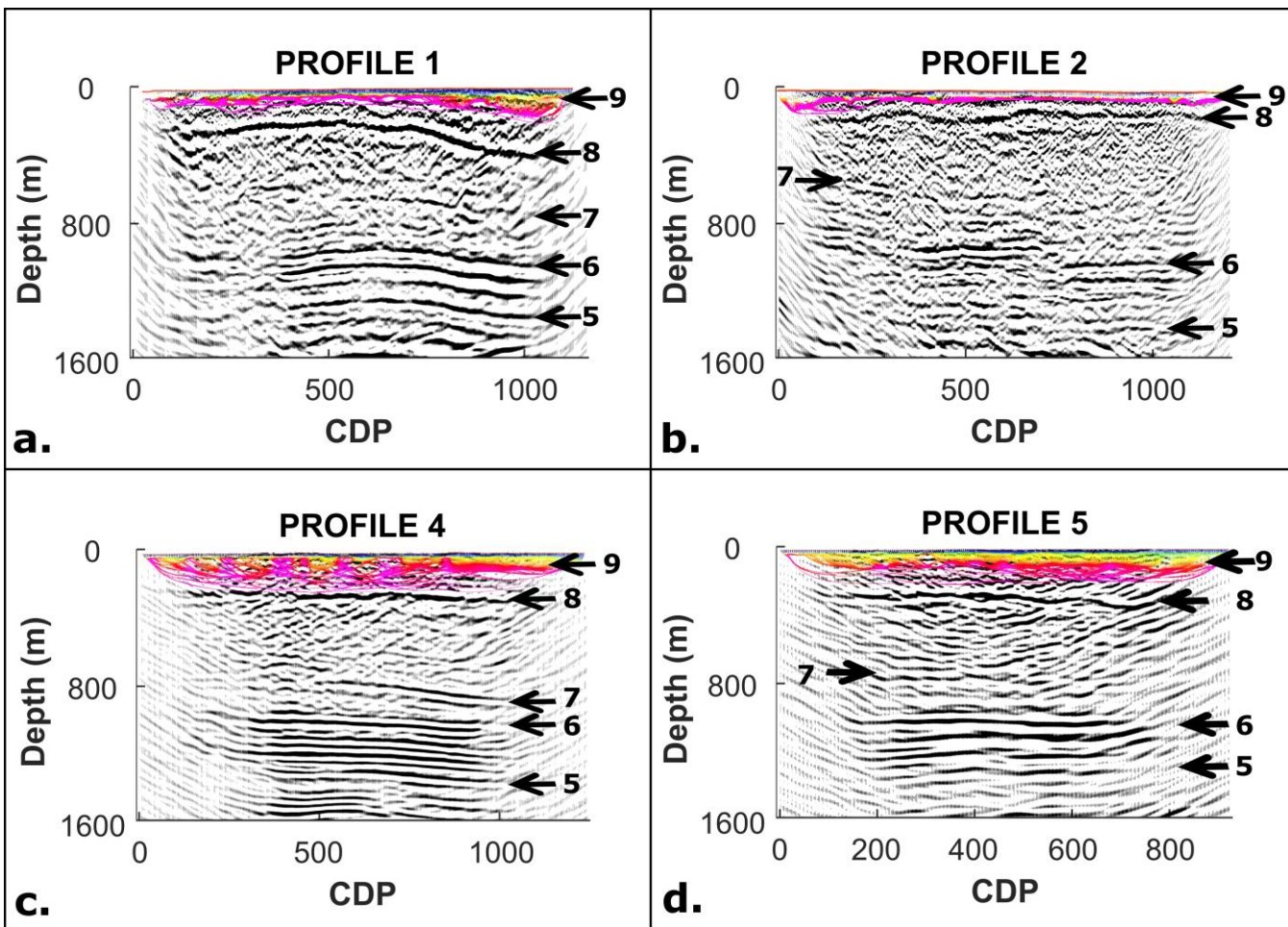

**Figure 10: A zoomed view of the upper 1600 meters from Figure 7, highlighting the near-surface Kalahari Group sediments and their seismic expression. This detailed section emphasizes the shallow stratigraphy and velocity variations within the cover**
**sediments, providing clearer insight into the near-surface geological framework. (5) the Ongeluk–Hotazel Formation contact; (6) the Hotazel–Mooidraai Formation contact; (7) the pre-Gamagara unconformity between the Transvaal and Keis Supergroups; (8) the Keis–Karoo Supergroup contact; and (9) the surface unconformity between the Karoo Supergroup and the overlying Kalahari Group. These reflectors highlight the internal basin architecture and stratigraphic continuity, with particular emphasis on the Hotazel Formation interval (reflections 5 and 6), which hosts the primary manganese mineralization targeted in this study.**

Interestingly, despite the substantial thicknesses of low-velocity Kalahari cover (between contact 9 and 8) in P4 and P5 (up to ~70 m), near-surface reflections are clearly imaged and show relatively high frequency content. This suggests that seismic energy was able to propagate effectively through the upper sand layers, a phenomenon attributed to the presence of lithified calcrete horizons. Calcretes, which form through pedogenic carbonate precipitation under semi-arid conditions, result in hardened crusts that increase the coherence and consolidation of the shallow subsurface (Bond, 1948; Verboom, 1974;
Lancaster, 2000). These lithified layers provide improved geophone coupling and allow for enhanced transmission of seismic

energy, thereby aiding in the imaging of deeper horizons. The effect is evident in the clarity of reflections within the shallow Kalahari strata observed in P4 and P5.

Nevertheless, the presence of thick unconsolidated aeolian sand in P4 and P5 also introduces complications. These layers act as waveguides for surface energy, resulting in strong ground roll and low-frequency guided waves that dominate the raw shot gathers. The guided wave energy, characterized by a dominant frequency range of 8–14 Hz, was particularly prominent in the interior farm profiles. This low-frequency noise is associated with the low shear-wave velocities of unconsolidated sand layers and is consistent with previous observations of near-surface seismic responses in arid regions (e.g., Xia et al., 1999).

The penetration depth of the low-frequency ground roll observed in the shot gathers can be approximated using the Rayleigh wave approximation Eq. (1):

$$h = \frac{\lambda}{2}, \qquad\qquad\qquad (1)$$

where $\lambda$ is the wavelength. Based on the measured moveout velocity of approximately 600 m/s for the ground roll and a dominant frequency range of 8–14 Hz, the estimated penetration depth ranges from ~21.5 m (at 14 Hz) to ~37.5 m (at 8 Hz). This estimate aligns well with the observed thickness of the shallow low-velocity zone in the tomographic model, supporting the interpretation that these guided waves are generated and sustained within the unconsolidated upper sand layer.

While calcrete horizons enhance coupling and allow signal transmission, the underlying unconsolidated units still lead to energy scattering and attenuation, reducing the resolution of deeper reflections, particularly below 4 km in P4 and P5. In contrast, the shallower aeolian sand cover along P1 and P2 allowed for greater high-frequency content and improved imaging continuity at depth.

Fig. 11 presents a three-dimensional visualization integrating the KPreSTM results overlain with the first vertical derivative (1VD) of the Total Magnetic Intensity (TMI) map (Fig. 11a) and the refraction tomographic velocity model (Fig. 11b) from all seismic profiles. The merged sections reveal a strong correlation of reflection events at profile intersections, confirming consistency in the structural interpretation across the survey grid. This spatial coherence enhances confidence in the interpolation of reflection continuity between profiles, allowing for improved delineation of subsurface stratigraphic and structural features. The integrated 3D visualization also highlights lateral continuity in near-surface velocity and reflectivity patterns that correspond well with major geological boundaries, offering a more comprehensive and coherent understanding

of the subsurface architecture. This interplay between the lithification state of the near surface, seismic energy propagation, and depth of investigation highlights the complexity of seismic imaging in semi-arid regions with variable surficial geology. Understanding the seismic response of such cover sediments is crucial for optimizing acquisition and processing strategies, especially in mineral exploration contexts like the Kalahari Manganese Field.

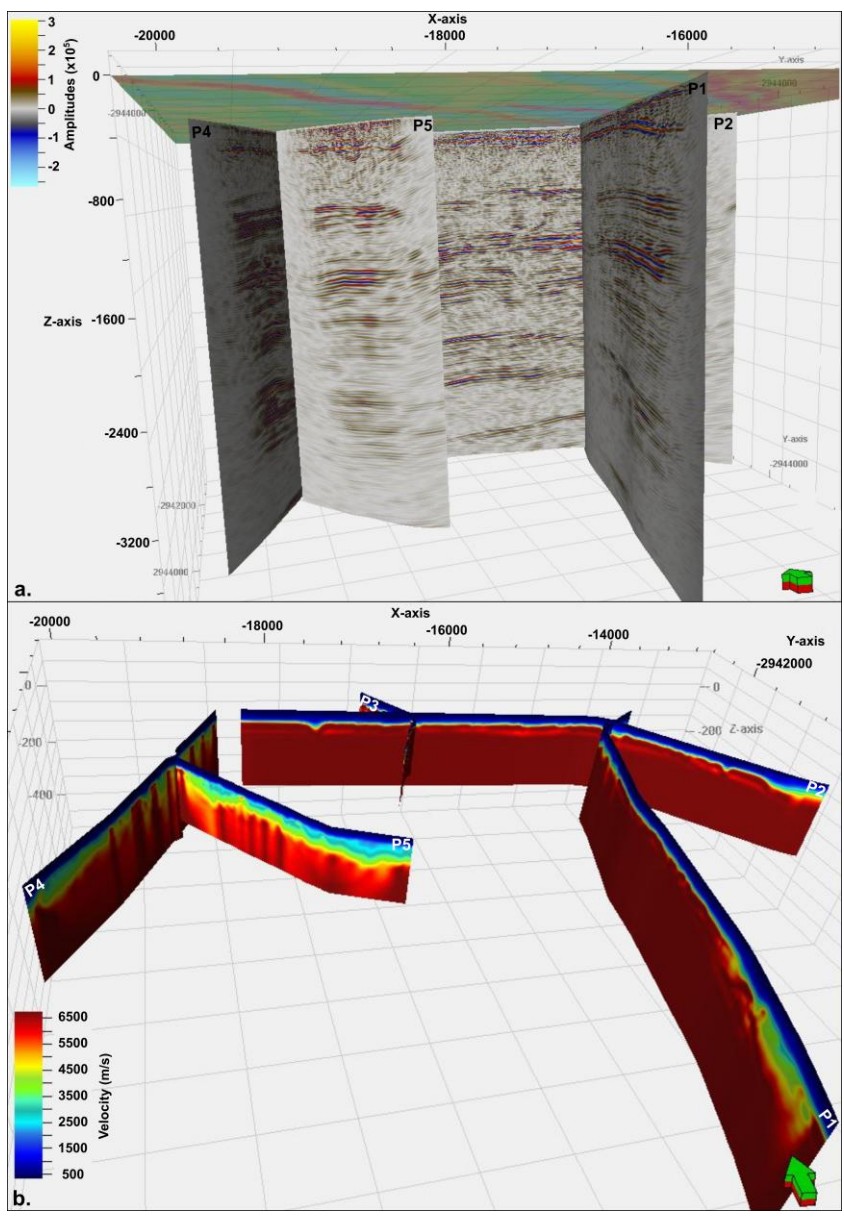

435

**Figure 11: Integrated 3D visualization of (a) Kirchhoff pre-stack time migration (KPreSTM) results overlain with the first vertical derivative (1VD) of the Total Magnetic Intensity (TMI) map, and (b) refraction tomographic velocity model from all seismic profiles.**

However, a poor signal-to-noise ratio is evident in some portions of the seismic sections, particularly in faulted zones and at the edges of the survey with low fold of coverage (blue box in Fig. 12). To enhance the continuity of the seismic reflectivity

440 and improve the accuracy of tracking seismic reflections associated with the mineralized zone (green box in Fig. 12) across all the sections, the cosine of phase attribute was computed. Fig. 12a-d show the comparison between the conventional seismic amplitude displays (Fig. 12a, c) and cosine of phase attribute displays (Fig. 12b, d). The cosine of phase attribute clearly reveals a degree of reflection continuity that is not well-defined by the amplitude displays. In particular, the cosine of phase attribute shows a better definition and delineation of the base of the Kalahari Group and Karoo Supergroup sediments, as well as the

445 continuity of the mineralized zones across the seismic sections (Fig.12c, d) when compared to the seismic amplitude displays (Fig. 12a, c) (green arrows in Fig. 12). The cosine of phase is independent of the amplitude and is thus less sensitive to noise and effective at defining the geometry and continuity of the reflections. Finally, the seismic attribute analysis and conventional seismic amplitude displays were used to pick the string reflections (red arrows in Fig. 12) associated with the mineralized zone within the Hotazel Formation across all seismic sections.

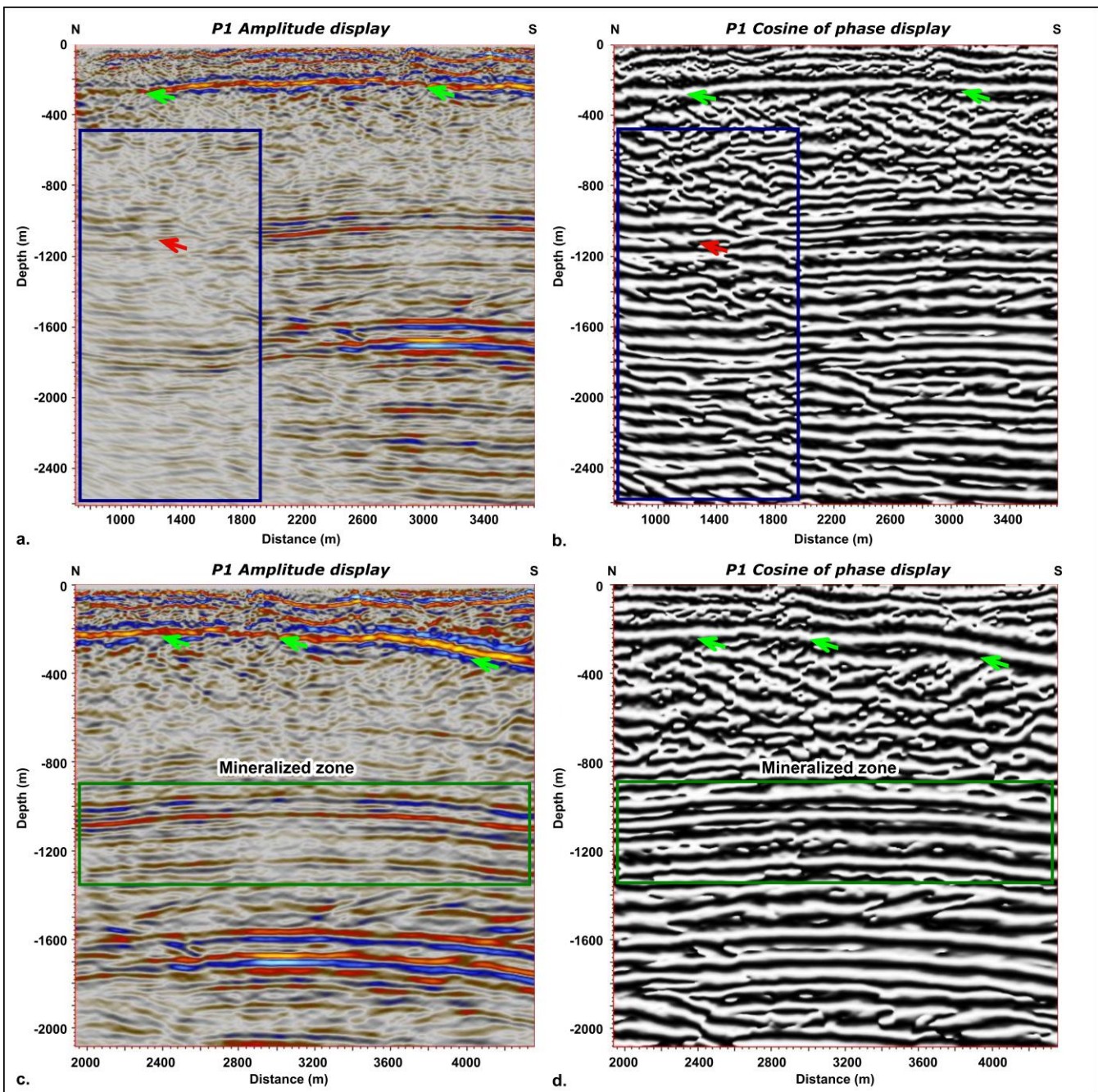

Figure 12: Comparison between conventional seismic amplitude and cosine of phase attribute displays. (a, c) Amplitude displays showing poor reflector continuity and signal degradation toward the survey edges and faulted zones. (b, d) Cosine of phase attribute displays revealing improved reflector continuity. Blue boxes indicate low fold and noisy zones; green boxes highlight the interpreted mineralized zones; green arrows emphasize enhanced reflection continuity; and red arrows mark improved reflections associated with the mineralized horizons.

Fig. 13 shows the 3D visualisation of all the seismic sections and picked horizons within the Hotazel Formation. The mineralized zone is defined by the top and bottom of the Hotazel Formation and has a total thickness of approximately 150 m. For future surveys in such environments, it is recommended to increase the maximum offset and overall spread length to enhance velocity analysis and improve depth imaging of deeper targets beneath the thick sand cover. Additionally, incorporating three-component (3C) geophones on selected profiles would allow the recording of S-wave information, aiding in the discrimination between surface waves and true reflection events. Furthermore, the surveys will provide a guide for future planned 3D seismic surveys in the region and subsequent deep exploration boreholes to intersect mineralisation at depths between 1.0 km and 1.5 km.

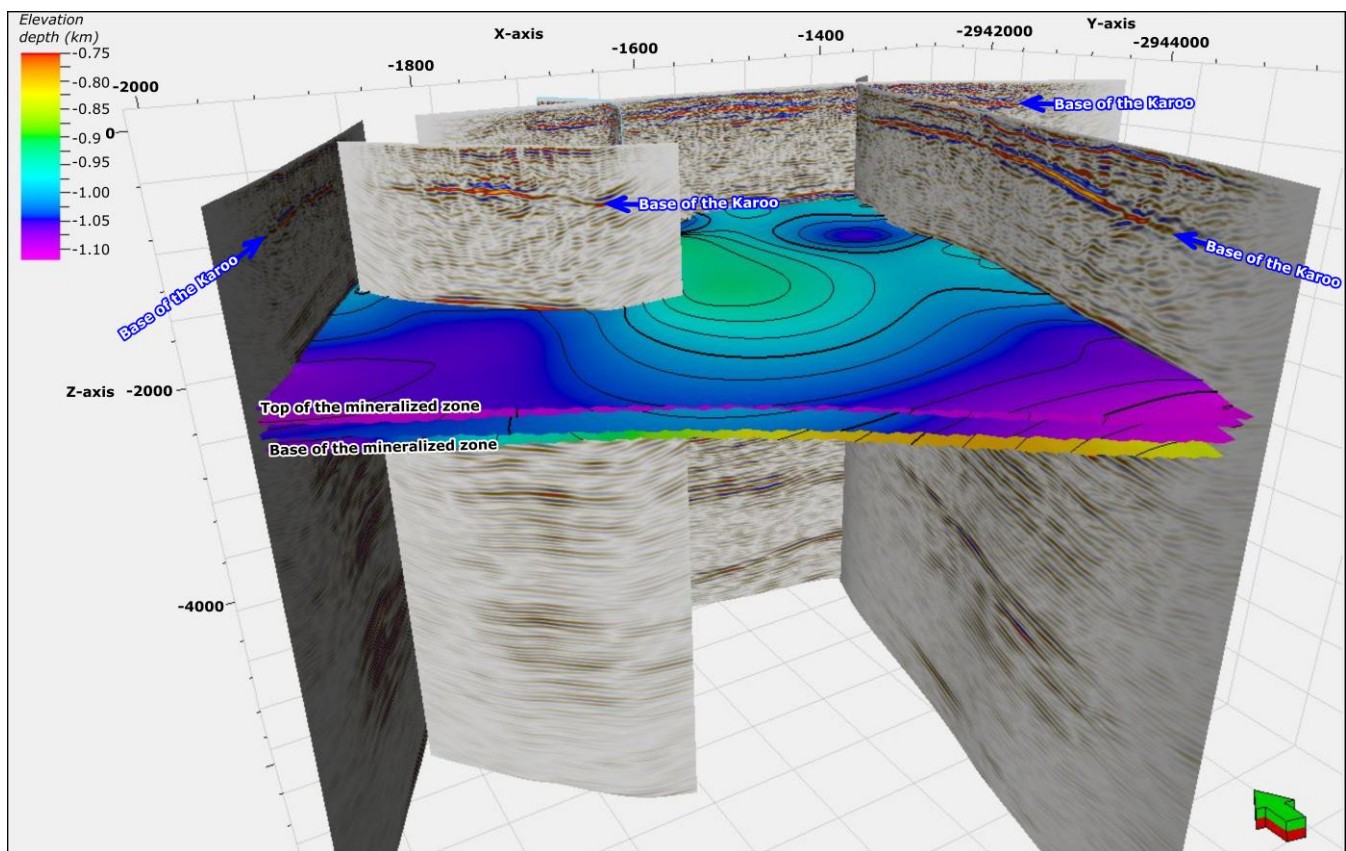

**Figure 13: 3D visualization of all seismic sections showing interpreted reflection horizons within the Hotazel Formation.**

## 5 Conclusion

A key outcome of the survey is the successful penetration of seismic energy through the thick Kalahari sand cover, long considered a major obstacle to geophysical exploration in the Kalahari Manganese Field (KMF). The Kalahari sand, which dominates the near-surface geology of the region, is predominantly composed of fine- to medium-grained, well-sorted, quartz-rich aeolian sands. Despite its loose and unconsolidated nature, several inherent properties of this material, including its low

clay content, high porosity, and dry condition, contribute to reduced seismic attenuation. These characteristics minimize both scattering and absorption of seismic energy, allowing for the propagation of coherent wavefields and improving signal-to-noise ratios at shallow depths.

This favourable wave propagation environment was further enhanced by the presence of indurated calcrete horizons. These
calcretized zones, developed through pedogenic carbonate accumulation, increase the stiffness and coherence of the shallow regolith, facilitating stronger seismic coupling and enabling deeper energy transmission. Together, the aeolian sands and calcrete crusts created a stratified yet acoustically efficient medium that allowed high-quality seismic imaging of underlying lithologies.

The choice of the 500 kg drop hammer as the seismic energy source was instrumental to the success of the survey. Mounted
on a compact and highly manoeuvrable Bobcat, the hammer could be efficiently deployed across diverse terrains, including deep sandy areas. Its mobility and relatively lightweight frame allowed for rapid repositioning and minimized the risk of vehicle immobilization in soft sand. In instances where it became stuck, the hammer assembly could be easily extracted using a 4x4 utility vehicle, further reducing delays and downtime.

Moreover, the high impulse force of the 500 kg drop hammer generated sufficient energy to image subsurface structures up to
depths exceeding 4 km, as evidenced by the continuity and resolution of the deep reflections. The simplicity and reliabilityof the source also contributed to the high signal fidelity observed in the shot gathers, making it ideal for stacking and processing. Low-frequency guided waves and ground roll generated within the unconsolidated sand layer were effectively managed during processing, allowing for improved imaging of deeper geological structures.

The effectiveness of the acquisition was further supported by the use of wireless nodes connected to 1C, 5 Hz geophones. The
nodes allowed for wireless, flexible deployment in rugged field conditions and, importantly, facilitated the burial of geophones beneath the loose surface sand. This not only improved seismic coupling by isolating sensors from wind-induced noise but also protected the cables from livestock interference, a frequent issue on operational farmlands.

Taken together, the combination of geologically informed acquisition design, lightweight high-energy source, and robust receiver technology enabled the successful acquisition of high-resolution 2D seismic data in a geologically complex and
logistically challenging region. These results demonstrate that, with careful adaptation of equipment and methodology, cost-effective seismic imaging can be achieved even in environments dominated by thick unconsolidated cover. The insights gained from this survey offer a valuable framework for future seismic exploration campaigns across the broader Kalahari Basin and other similar sediment-covered mineral provinces in southern Africa.

**Author contribution**

Mpofana Sihoyiya carried out data curation, investigation, methodology, visualization, and original draft preparation. Musa S. D. Manzi provided conceptualization, funding acquisition, project administration, resources, supervision, and contributed to

review and editing. Ian James performed formal analysis, software, and contributed to review and editing. Michael Westgate provided validation and contributed to review and editing.

**Acknowledgements**

The authors gratefully acknowledge African Exploration Mining and Finance Corporation (AEMFC) and DSTI-NRF Centre of Excellence (CoE) for Integrated Mineral and Energy Resource Analysis (CIMERA) for funding this research project. We extend our sincere thanks to the Wits Seismic Research Team from the School of Geosciences, University of the Witwatersrand (South Africa) for their invaluable assistance during seismic data acquisition. Their field support and technical expertise contributed significantly to the success of this study. Globe Claritas and SLB are thanked for providing the academic licences

for processing and interpretation software packages, respectively. We also thank the reviewers and the Associate Editor for providing constructive feedback that improved the quality of this manuscript.

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
