# Peer review of "2D reflection seismic surveys to delineate manganese mineralisation beneath the thick Kalahari and Karoo cover in the Griqualand West Basin, South Africa"

_EGUsphere, 2025_

## Referee Comment (RC2)

[referee-annotated manuscript omitted]

---

## Author Response (AR1)

Private Bag 3, Wits 2050, Johannesburg, South Africa – Tel: +27 79 577 629 4

Email: 1284606@students.wits.ac.za

**School of Geosciences**

28 October 2025

Dear Prof. Juhlin

We are pleased to submit the revised manuscript eguphere-2025-3117 entitled "**Reflection seismic imaging of the manganese mineralisation in the Griqualand West Basin, South Africa**" for publication. We appreciate and have addressed the constructive comments from the Editor, the Assistant Editor, and the reviewers. Below we reviewed the comments and incorporated them into the manuscript, please note that the original comments our responses are printed in blue for reviewer 1, and green for reviewer 2. The line numbers are referred to in the file Sihoyiya et al._track_changes.docx. All the changes to the manuscript are tracked in the file Sihoyiya et al._track_changes.docx. A clean version of the manuscript is also provided in the file Sihoyiya et al._clean.docx.

We believe that the manuscript is now much improved, and we hope that you find it acceptable for publication.

Sincerely,

Mpofana Sihoyiya and co-authors

**Commented:** Main geological interpretation results (Nine reflectors) were presented in the paper titled "Reflection seismic imaging of the manganese mineralisation in the Griqualand West Basin, South Africa" by Joggee et al. (2025), the first and the second authors are also co-authors of the

published paper. The difference of the manuscript and their prior paper and new findings must be presented in the manuscript.

- Corrected and updated on Line 95 of Sihoyiya et al. 2025_Track_Changes.

**Commented:** Authors stated a lot about data acquisition method to obtain high-quality and - resolution seismic data, not reflection seismic imaging in the conclusion section. They do not correspond to the title of the manuscript. Authors should focus on what you want to present in the manuscript.

- Corrected and updated on Line 1 of Sihoyiya et al. 2025_Track_Changes.

**Commented:** Line 65, Figure 1a is not clear. It is important to give regional and local geological maps to help readers understand geological background. A typical lithostratigraphic section is preferable. And the location and layout of seismic survey lines do not need to be shown in Figure 1b, there are displayed in Figure 2.

- Corrected and updated on Line 71 of Sihoyiya et al. 2025_Track_Changes.

**Commented:** Line 181, the recording time of raw shot gather in Figure 5c is 1000 ms, not same with other shot gathers. Is it right? Reflections should be marked in Figure 5c-5e.
- Corrected and updated on Line 228 of Sihoyiya et al. 2025_Track_Changes.

**Commented:** Line 186, main data processing methods and key parameters should be given in a table and examples of raw shot gathers dominated by reflection waves or with clear reflections waves should be shown after data processing in the processing section.

- Corrected and updated on Line 262 and 228 of Sihoyiya et al. 2025_Track_Changes.

**Commented:** Line 188, pre-stack migration was applied in this study, but I cannot see any pre-stack migration results in the manuscript. The Kirchhoff post-stack time-migrated seismic sections is stated in the caption of Figure 7. What kind of migration methods did you use?
- Corrected and updated on Line 389 of Sihoyiya et al. 2025_Track_Changes.

**Commented:** Line 199, You may consider to focus on near-surface structure with refraction tomography results.

- Corrected and updated on Line 305 of Sihoyiya et al. 2025_Track_Changes.

**Commented:** Line 251-270, nine reflectors were interpreted through reflection seismic profiles. However, reflectors numbered 1 and 2 do not agree with the results of your prior results (Joggee et al., 2025). I am wondering if there are your new results.

- Corrected and updated on Line 344 and 347 of Sihoyiya et al. 2025_Track_Changes.

**Commented:** Line 274, "stud" should be "study"?

- Corrected and updated on Line 369 of Sihoyiya et al. 2025_Track_Changes.

**Commented:** Line 292, I don't understand why you didn't provide profile 3 result.

- Thank you for pointing this out. Profile 3 was omitted due to poor data quality, which made reliable interpretation difficult.

**Commented:** Line 320, profile 3 was also missed in Figure 8.

- Thank you for pointing this out. Profile 3 was omitted due to poor data quality, which made reliable interpretation difficult.

**Commented:** Overall, the manuscript is clearly written and mostly well structured (see comments in the pdf). However, its novelty appears rather limited compared to Jogee et al. (2025). Most of the processing and interpretation steps are essentially identical to the earlier publication. The only substantial addition seems to be the tomographic component, while even the perspective 3D view and horizon picking were already shown previously — yet these are not included here. To strengthen the contribution, I recommend a more explicit comparison with the earlier work and a clearer demonstration of what is new in this study.

- Corrected and updated on Line 95 of Sihoyiya et al. 2025_Track_Changes.

**Commented:** Please add a table summarizing the acquisition parameters.

- Added on Line 172 of Sihoyiya et al. 2025_Track_Changes.

**Commented:** Please include examples of the seismic data both before and after processing. This would allow readers to better evaluate the impact of your workflow (see also the comments in the pdf ).

- Added on Line 208 of Sihoyiya et al. 2025_Track_Changes.

**Commented:** Provide a more detailed explanation of how guided waves were handled during processing.

- Added on Line 242 of Sihoyiya et al. 2025_Track_Changes.

**Commented:** Currently you are showing less information compared to Jogee et al. (2025.

- We have added the focus of the current study on Line 95 of Sihoyiya et al. 2025_Track_Changes, emphasized the new contributions, and provided additional figures or supplementary material on Sihoyiya et al. 2025_Track_Changes.

**Commented:** When describing sediment thicknesses across the profiles, consider including a map of thickness distribution. This would help clarify where the sedimentary cover (low-velocity zone) is thickest and how it may impact image quality.

- Thank you for the valuable comment. To the authors' knowledge, no sediment thickness map has been published for the study area. The interpretation of sediment thickness in this work is constrained using results and geological information from previous studies conducted in the region (Reimold et al., 2002; Tinker et al., 2002; Tsikos and Moore, 1997; Beukes et al., 2016; Westgate et al., 2020).

**Commented:** Profile 3 appears to be missing

- Thank you for pointing this out. Profile 3 was omitted due to poor data quality, which made reliable interpretation difficult.

**Commented:** Ensure that the profiles are displayed with the correct aspect ratio (Profiles 1 and 2 are twice as long as Profile 5.

- Corrected and updated on Line 388 of Sihoyiya et al. 2025_Track_Changes.

**Commented:** Clarify whether different CDP bin sizes were used across the profiles.

- Clarified on Line 239 of Sihoyiya et al. 2025_Track_Changes.

**Commented:** You provide a detailed overview of the expected boundaries. Have you computed a synthetic seismogram from the series of reflection coefficients? This would allow for a direct comparison with the seismic images and could significantly strengthen the interpretation.

- Added on Line 406 of Sihoyiya et al. 2025_Track_Changes.

**Commented:** Why is no 3D interpretation or perspective view included here, given that one was already presented in Jogee et al. (2025).

- Added on Line 475 and 506 of Sihoyiya et al. 2025_Track_Changes.

**Commented:** Please add interpretation figures for each profile (highlighting the main geological units, etc.), similar to Profile 1 in Figure 12 of Jogee et al. (2025)

- Thank you very much for the constructive comment. We appreciate the suggestion to include interpretation figures for each profile. However, the main focus of this manuscript is on the methodological and processing aspects of the data. The detailed geological interpretations, including annotated figures, have already been comprehensively presented in Jogee et al. (2025). To avoid repetition, we have therefore not included these figures here.

**Commented:** Line 13: these technical details should not appear in the abstract

- Corrected and updated on Line 14 of Sihoyiya et al. 2025_Track_Changes.

**Commented:** Line 25: Mn

- Corrected and updated on Line 30 of Sihoyiya et al. 2025_Track_Changes.

**Commented:** Line 27: Mn

- Corrected and updated on Line 33 of Sihoyiya et al. 2025_Track_Changes.

**Commented:** Line 32: Mn

- Corrected and updated on Line 37 of Sihoyiya et al. 2025_Track_Changes.

**Commented:** Line 33: what kind of geophyical surveys? reflection seismics?

- Corrected and updated on Line 38 of Sihoyiya et al. 2025_Track_Changes.

**Commented:** Line 55: This sentence stands somewhat alone.

- Corrected and updated on Line 60 of Sihoyiya et al. 2025_Track_Changes.

**Commented:** Line 57: what do you mean by carfully planned?

- Corrected and updated on Line 62 of Sihoyiya et al. 2025_Track_Changes.

**Commented:** Line 58: why is the survey cost effective?

- Thank you for the comment. The survey is considered cost-effective because it employed a small and portable seismic source (GPEG500), which significantly reduced mobilisation, operational, and acquisition costs compared to conventional large-scale vibroseis or explosive sources, while still achieving sufficient energy penetration and data quality for imaging the target manganese-bearing horizons.

**Commented:** Line 62: number of Geophones? Geophone spacing?

- Thank you for the comment. The number of geophones and their spacing are provided in the acquisition summary table, as these parameters vary between profiles due to differences in line length, terrain conditions, and acquisition objectives.

**Commented:** Line 65: add a box showing the location of b.

- Corrected and updated on Line 71 of Sihoyiya et al. 2025_Track_Changes.

**Commented:** Line 72: UAV was mentioned already in the sentence before

- Corrected and updated on Line 83 of Sihoyiya et al. 2025_Track_Changes.

**Commented:** Line 95: varying from ? to ?

which order of magnitude?

- Corrected and updated on Line 115 of Sihoyiya et al. 2025_Track_Changes.

**Commented:** Line 99: again, how thick are these layers? Can we expect that they can be resolved using refleciton seismics?

- Thank you for the comment. The manganese-bearing layers in the study area are relatively thin, ranging between 5 and 8 m in thickness, while they can reach up to approximately 45 m in areas south of the study site.

**Commented:** Line 107 – 118: I think this should not be part of the geology section. You can move this to the seismic section instead.

- Thank you for the comment. The section provides important geological context relevant to the seismic interpretation and is therefore briefly introduced here to establish the geological framework. However, we acknowledge your suggestion and will streamline this part in the geology section, while expanding the discussion of its seismic relevance in the seismic interpretation section for better structural coherence.

**Commented:** Line 138: delete high resolution

- Corrected and updated on Line 160 of Sihoyiya et al. 2025_Track_Changes.

**Commented:** Line 169: noise

- Corrected and updated on Line 187 of Sihoyiya et al. 2025_Track_Changes.

**Commented:** Line 163: what is this stripe pattern?

- Corrected and updated on Line 190 of Sihoyiya et al. 2025_Track_Changes.

**Commented:** Line 163: Typo : ground roll, guided waves

- Corrected and updated on Line 190 of Sihoyiya et al. 2025_Track_Changes.

**Commented:** Line 182: there is not shot 98 figure 1.

It would also be good to highlight the shown shots in fig. 1. And additionally show shots at the same locations for the crossing lines.

- Corrected and updated on Line 228 of Sihoyiya et al. 2025_Track_Changes.

**Commented:** Line 195: What is the reason for 10Hz lower boundary?

- Thank you for the comment. The lower boundary of 10 Hz was chosen because most of the raw shot gathers are dominated by low-frequency ground roll energy in the 8–14 Hz range. Setting the lower cutoff at 10 Hz was intended to mitigate the influence of ground roll energy.

**Commented:** Line 218: new paragraph.

- Corrected and updated on Line 283 of Sihoyiya et al. 2025_Track_Changes.

**Commented:** Line 240: Is the initial model still part of the results? can you restrict the velocity models to the ray-covered parts?!

- Thank you for the helpful comment. The tomographic velocity models presented in Figures 7 and 9 are indeed restricted to the ray path–covered zones. However, when shown as standalone models, limiting them strictly to these zones reduces their visual clarity and continuity.

**Commented:** Line 240: It would also be good to compare the intersections of the profiles in more detail, eg using velocity vs. Depth profiles at these locations

- Added on Line 475 of Sihoyiya et al. 2025_Track_Changes.

**Commented:** Line 286: which numbers?

- Corrected and updated on Line 381 of Sihoyiya et al. 2025_Track_Changes.

**Commented:** Line 308 – 313: move this to 3.3

- Corrected and updated on Line 419 of Sihoyiya et al. 2025_Track_Changes.

**Commented:** Line 355: how would you optimize a future survey in this region, based on your experience with these data?

- Corrected and updated on Line 500 of Sihoyiya et al. 2025_Track_Changes.

**Commented:** Line 369 – 375: I cannot agree with your argument. Both, explosives and conventional vibroseis trucks are not cauing problems when operating on sand....

- Corrected and updated on Line 529 and 531 of Sihoyiya et al. 2025_Track_Changes.

**Commented:** Line 379: please add examples to demonstrate the repeatability your mentioned enhanced signal-to-noise ratio

- Updated on Line 537 of Sihoyiya et al. 2025_Track_Changes.

---

## Author Response (AR2)

Private Bag 3, Wits 2050, Johannesburg, South Africa – Tel: +27 79 577 629 4

Email: 1284606@students.wits.ac.za

**School of Geosciences**

28 October 2025

Dear Prof. Juhlin

We are pleased to submit the revised manuscript eguphere-2025-3117 entitled "**Reflection seismic imaging of the manganese mineralisation in the Griqualand West Basin, South Africa**" for publication. We appreciate and have addressed the constructive comments from the Editor, the Assistant Editor, and the reviewers. Below we reviewed the comments and incorporated them into the manuscript, please note that the original comments our responses are printed in blue for reviewer 1, and green for reviewer 2. The line numbers are referred to in the file Sihoyiya et al._track_changes.docx. All the changes to the manuscript are tracked in the file Sihoyiya et al._track_changes.docx. A clean version of the manuscript is also provided in the file Sihoyiya et al._clean.docx.

We believe that the manuscript is now much improved, and we hope that you find it acceptable for publication.

Sincerely,

Mpofana Sihoyiya and co-authors

**Commented:** Main geological interpretation results (Nine reflectors) were presented in the paper titled "Reflection seismic imaging of the manganese mineralisation in the Griqualand West Basin, South Africa" by Joggee et al. (2025), the first and the second authors are also co-authors of the

published paper. The difference of the manuscript and their prior paper and new findings must be presented in the manuscript.

We thank the reviewer for this valuable comment. This paper is distinct from, and improves upon Jogee et al. (2025) on a number of points. The major ones are:

- Near-surface characterization: Integration of both reflected and refracted wavefield energy to provide a comprehensive understanding of the shallow subsurface architecture.
- Numerical seismic simulations: Application of modeling to optimize geological interpretation in complex geological settings such as the Griqualand West Basin.
- Migration comparison: Presentation and comparison of pre-stack time and depth Kirchhoff migration results to evaluate their relative performance in imaging manganese-bearing stratigraphic horizons and geological contacts.
- Seismic attribute analysis: Computation of seismic attributes (e.g., cosine of phase) to enhance the detection of manganese-mineralized zones and comparison with traditional seismic amplitude displays.

These points are implicit in the paper, however we have added a paragraph in the introduction (Line 95 of Sihoyiya et al. 2025_Track_Changes) explicitly summarizing the new contributions of this manuscript.

**Commented:** Authors stated a lot about data acquisition method to obtain high-quality and -resolution seismic data, not reflection seismic imaging in the conclusion section. They do not correspond to the title of the manuscript. Authors should focus on what you want to present in the manuscript.

We appreciate the reviewer's helpful observation. To ensure consistency between the manuscript content and its title (Line 1 of Sihoyiya et al. 2025_Track_Changes), we have made the following change:

- Revised title: "2D reflection seismic surveys to delineate manganese mineralisation beneath the thick Kalahari and Karoo cover in the Griqualand West Basin, South Africa."

This revised title better reflects the main focus of the manuscript, which emphasizes the application of 2D reflection seismic methods for delineating manganese mineralisation rather than solely describing acquisition procedures.

**Commented:** Line 65, Figure 1a is not clear. It is important to give regional and local geological maps to help readers understand geological background. A typical lithostratigraphic section is preferable. And the location and layout of seismic survey lines do not need to be shown in Figure 1b, there are displayed in Figure 2.

We thank the reviewer for this valuable suggestion. We have revised Figure 1 (Line 71 of Sihoyiya et al. 2025_Track_Changes) to improve clarity and geological context. The updated figure now includes:

- (a) A regional geological map of the Transvaal Supergroup and lower Keis Supergroup, with the study area indicated by a black square.
- (b) A lithostratigraphic section from the Sishen area in the southwest to the Kalahari Manganese Field (KMF) in the northeast, illustrating the stratigraphic relationships among the Ongeluk, Hotazel, and Mooidraai formations (modified after Hongjun et al., 2022; Jogee et al., 2025).
- (c) A regional geological map of the study area showing the distribution of the main lithostratigraphic units and the extent of the Kalahari cover (modified after Jogee et al., 2025).

Additionally, we removed the seismic survey line locations from Figure 1, as these are already shown in Figure 2.

**Commented:** Line 181, the recording time of raw shot gather in Figure 5c is 1000 ms, not same with other shot gathers. Is it right? Reflections should be marked in Figure 5c-5e.

We thank the reviewer for this observation. Figure 5c represents results from a shorter profile (Profile 3) compared to the other profiles. To enhance clarity, only the upper 1000 ms was originally displayed for better visualization. To maintain constancy we have made the following changes:

- Standardized recording time: All example shot gather images have been updated to have the same final recording time for consistency (see Line 228 in Sihoyiya et al., 2025_Track_Changes).
- Reflections marked: The reflection events have been clearly marked in Figures 5f –5j.

**Commented:** Line 186, main data processing methods and key parameters should be given in a table and examples of raw shot gathers dominated by reflection waves or with clear reflections waves should be shown after data processing in the processing section.

We appreciate the reviewer's constructive suggestion, we have made the following updates:

- Processing parameters table added: A detailed table summarizing the main data processing steps and key parameters has been included (see Line 262 in Sihoyiya et al., 2025_Track_Changes).
- Processed example shot gathers added: Representative examples of shot gathers showing clear reflection events after processing have been included in Figure 5 of the processing section to illustrate data quality improvement (see Line 262 of Sihoyiya et al. 2025_Track_Changes).

**Commented:** Line 188, pre-stack migration was applied in this study, but I cannot see any pre-stack migration results in the manuscript. The Kirchhoff post-stack time-migrated seismic sections is stated in the caption of Figure 7. What kind of migration methods did you use?

We thank the reviewer for pointing out this important clarification. In this study, multiple migration methods were tested; however, we present the results from the Kirchhoff pre-stack time migration (PreSTM) and Kirchhoff pre-stack depth migration (PreSDM) methods, which were applied in this work. The reference to "Kirchhoff post-stack time migration" in the caption of Figure 7 was a typographical error and has been corrected in the revised manuscript as follows:

- Updated figure captions and text: The captions of Figures 7 and 8, as well as the relevant parts of the Processing and Imaging section, have been revised to correctly reflect the use of pre-stack migration methods (see Lines 326 and 389 of Sihoyiya et al. 2025_Track_Changes).

- Included comparison results: Results from both PreSTM and PreSDM sections are now presented and discussed to evaluate their relative performance in imaging the manganese-bearing horizons and structural features.

**Commented:** Line 199, You may consider to focus on near-surface structure with refraction tomography results.

We thank the reviewer for this valuable suggestion. We have expanded the interpretation of the refraction tomographic velocity model to include the near-surface structure. Specifically, we interpreted the low-velocity zones (600–4000 m/s) as Kalahari Group sediments, comprising aeolian sands and gravels. The corresponding revisions have been incorporated into the "Reflection Tomography" section (Line 305 of Sihoyiya et al., 2025_Track_Changes).

**Commented:** Line 251-270, nine reflectors were interpreted through reflection seismic profiles. However, reflectors numbered 1 and 2 do not agree with the results of your prior results (Joggee et al., 2025). I am wondering if there are your new results.

We thank the reviewer for noting this discrepancy. The inconsistency was due to an error in labeling. The reflector numbering has been corrected to ensure consistency with Jogee et al. (2025). Specifically, the contacts have been updated as follows:

- Updated reflector contacts: The interpretation of reflectors 1 and 2 has been corrected to align with Joggee et al. (2025):
- Reflector 1: Ventersdorp Supergroup – Campbellrand and Schmidtsdrif contact
- Reflector 2: Campbellrand and Schmidtsdrif – Asbestos Hills and Koegas contact

This correction ensures consistency with our prior published results (see Lines 344 and 347 of Sihoyiya et al. 2025_Track_Changes).

**Commented:** Line 274, "stud" should be "study"?

We thank the reviewer for catching this typographical error.

- Correction made: The word "stud" has been corrected to "study" (see Line 369 in Sihoyiya et al., 2025_Track_Changes).

**Commented:** Line 292, I don't understand why you didn't provide profile 3 result.

We thank the reviewer for raising this point. Profile 3 was omitted from the manuscript due to poor data quality, which prevented reliable interpretation and meaningful visualization of subsurface structures.

**Commented:** Line 320, profile 3 was also missed in Figure 8.

We thank the reviewer for raising this point. Profile 3 was omitted from the manuscript due to poor data quality, which prevented reliable interpretation and meaningful visualization of subsurface structures.

**Commented:** Overall, the manuscript is clearly written and mostly well structured (see comments in the pdf). However, its novelty appears rather limited compared to Jogee et al. (2025). Most of the processing and interpretation steps are essentially identical to the earlier publication. The only substantial addition seems to be the tomographic component, while even the perspective 3D view and horizon picking were already shown previously — yet these are not included here. To strengthen the contribution, I recommend a more explicit comparison with the earlier work and a clearer demonstration of what is new in this study.

We thank the reviewer for this valuable feedback. To strengthen the manuscript and clarify its novelty, we have incorporated the following main additions:

- Integration of reflection and refraction wavefield energy for near-surface characterization.
- Use of numerical seismic simulations to guide interpretation.
- Comparison of pre-stack time and depth migration results to evaluate imaging performance of manganese-bearing horizons.
- Application of seismic attributes (e.g., cosine of phase) to enhance detection of mineralized zones.

These revisions aim to clearly demonstrate the added value of the current study relative to previous work. A paragraph has been added on Line 95 of Sihoyiya et al., 2025_Track_Changes to highlight the new contributions.

**Commented:** Please add a table summarizing the acquisition parameters.

- We thank the reviewer for the suggestion.

- Acquisition parameters table added: A table summarizing all acquisition parameters has been included in the manuscript (see Line 172 in Sihoyiya et al., 2025_Track_Changes).

**Commented:** Please include examples of the seismic data both before and after processing. This would allow readers to better evaluate the impact of your workflow (see also the comments in the pdf ).

We thank the reviewer for the helpful suggestion. A table summarizing all acquisition parameters has been added to the manuscript for clarity and completeness (see Line 172 in Sihoyiya et al., 2025_Track_Changes).

**Commented:** Provide a more detailed explanation of how guided waves were handled during processing.

We thank the reviewer for this valuable suggestion. Guided waves were removed using a radial trace transform filter, where the guided-wave energy was isolated in the radial trace domain, transformed back to the spatial–temporal domain, and subtracted from the data. In addition, FK filtering was applied to further attenuate any remaining guided-wave energy.

- A detailed explanation of how guided waves were identified and removed during data processing has been added (see Line 242 in Sihoyiya et al., 2025_Track_Changes).

**Commented:** Currently you are showing less information compared to Jogee et al. (2025.

We thank the reviewer for this valuable feedback. To strengthen the manuscript and clarify its novelty, we have incorporated the following main additions:

- Integration of reflection and refraction wavefield energy for near-surface characterization.
- Use of numerical seismic simulations to guide interpretation.
- Comparison of pre-stack time and depth migration results to evaluate imaging performance of manganese-bearing horizons.
- Application of seismic attributes (e.g., cosine of phase) to enhance detection of mineralized zones.

These revisions aim to clearly demonstrate the added value of the current study relative to previous work. A paragraph has been added on Line 95 of Sihoyiya et al., 2025_Track_Changes to highlight the new contributions.

**Commented:** When describing sediment thicknesses across the profiles, consider including a map of thickness distribution. This would help clarify where the sedimentary cover (low-velocity zone) is thickest and how it may impact image quality.

Thank you for the valuable comment. To the authors' knowledge, no sediment thickness map has been published for the study area. The interpretation of sediment thickness in this work is constrained using results and geological information from previous studies conducted in the region (Reimold et al., 2002; Tinker et al., 2002; Tsikos and Moore, 1997; Beukes et al., 2016; Westgate et al., 2020).

**Commented:** Profile 3 appears to be missing

Thank you for pointing this out. Profile 3 was omitted due to poor data quality, which made reliable interpretation difficult.

**Commented:** Ensure that the profiles are displayed with the correct aspect ratio (Profiles 1 and 2 are twice as long as Profile 5.

We thank the reviewer for this comment. The profile displays are exaggerated for visualization purposes. The seismic profiles in Figure 8 have now been revised as follows:

- Corrected aspect ratio: The seismic profiles in Figure 8 have been revised to display with the correct aspect ratio, ensuring consistency across all profiles (see Line 388 in Sihoyiya et al., 2025_Track_Changes).

**Commented:** Clarify whether different CDP bin sizes were used across the profiles.

We thank the reviewer for this comment. The CDP bin size along each profile was defined as half of the receiver spacing. Therefore, different bin sizes were used across profiles because the receiver spacing varied between them.

- CDP bin size clarification: Details of CDP bin construction have been added on Line 239 of Sihoyiya et al., 2025_Track_Changes.

**Commented:** You provide a detailed overview of the expected boundaries. Have you computed a synthetic seismogram from the series of reflection coefficients? This would allow for a direct comparison with the seismic images and could significantly strengthen the interpretation.

We thank the reviewer for this valuable suggestion. A 1D synthetic seismogram was created to test the source of reflectivity and to constrain the interpretation. This comparison allows for a more robust correlation between the modeled reflection series and the observed seismic images. The details have been added on Line 406 of Sihoyiya et al., 2025_Track_Changes.

**Commented:** Why is no 3D interpretation or perspective view included here, given that one was already presented in Jogee et al. (2025).

We thank the reviewer for this valuable suggestion. Initially, we had not included a 3D perspective view to avoid repetition; however, we recognize that including it provides additional context and continuity with previous work. The revised manuscript now includes the following:

- 3D interpretation added: Integrated 3D visualizations have been included in the manuscript (see Lines 475 and 506 in Sihoyiya et al., 2025_Track_Changes):
- Figure 11 shows 3D visualization of Kirchhoff pre-stack time migration (KPreSTM) results overlain with the first vertical derivative (1VD) of the Total Magnetic Intensity (TMI) map and Refraction tomographic velocity model from all seismic profiles.
- Additional 3D visualization: All seismic sections showing interpreted reflection horizons within the Hotazel Formation.

These additions provide a clearer spatial understanding of the subsurface and highlight the integrated interpretation of reflection and refraction data.

**Commented:** Please add interpretation figures for each profile (highlighting the main geological units, etc.), similar to Profile 1 in Figure 12 of Jogee et al. (2025)

Thank you very much for the constructive comment. We appreciate the suggestion to include interpretation figures for each profile. However, the main focus of this manuscript is on the methodological and processing aspects of the data. The detailed geological interpretations, including annotated figures, have already been comprehensively presented in Jogee et al. (2025). To avoid repetition, we have therefore not included these figures here.

**Commented:** Line 13: these technical details should not appear in the abstract

We thank the reviewer for this comment. The following are the technical details removed from abstract:

- The phrase "yielding nearly five million seismic traces" has been removed to streamline the abstract (see Line 14 in Sihoyiya et al., 2025_Track_Changes).

**Commented:** Line 25: Mn

We thank the reviewer for this comment. The following correction has been made in the manuscript:

- Abbreviation correction: Manganese has been abbreviated to Mn (see Line 30 in Sihoyiya et al., 2025_Track_Changes).

**Commented:** Line 27: Mn

We thank the reviewer for this comment. The following correction has been made in the manuscript:

- Abbreviation correction: Manganese has been abbreviated to Mn (see Line 33 in Sihoyiya et al., 2025_Track_Changes).

**Commented:** Line 32: Mn

We thank the reviewer for this comment. The following correction has been made in the manuscript:

- Abbreviation correction: Manganese has been abbreviated to Mn (see Line 37 in Sihoyiya et al., 2025_Track_Changes).

**Commented:** Line 33: what kind of geophyical surveys? reflection seismics?

We thank the reviewer for this comment. The geophysical survey referred to is ground magnetics, and this has been updated in the manuscript (see Line 38 in Sihoyiya et al., 2025_Track_Changes).

**Commented:** Line 55: This sentence stands somewhat alone.

We thank the reviewer for this comment. The standalone sentence has been removed from the manuscript (see Line 60 in Sihoyiya et al., 2025_Track_Changes).

**Commented:** Line 57: what do you mean by carfully planned?

We thank the reviewer for this comment. The term "carefully planned" has been removed to improve clarity (see Line 62 in Sihoyiya et al., 2025_Track_Changes).

**Commented:** Line 58: why is the survey cost effective?

Thank you for the comment. The survey is considered cost-effective because it employed a small and portable seismic source (GPEG500), which significantly reduced mobilisation, operational, and acquisition costs compared to conventional large-scale vibroseis or explosive sources, while still achieving sufficient energy penetration and data quality for imaging the target manganese-bearing horizons.

**Commented:** Line 62: number of Geophones? Geophone spacing?

Thank you for the comment. The number of geophones and their spacing are provided in the acquisition summary table, as these parameters vary between profiles due to differences in line length, terrain conditions, and acquisition objectives.

**Commented:** Line 65: add a box showing the location of b.

We thank the reviewer for this valuable suggestion. The figure has been updated on Line 71 of Sihoyiya et al., 2025_Track_Changes to improve clarity and geological context.

**Commented:** Line 72: UAV was mentioned already in the sentence before

We thank the reviewer for this comment. The phrase "mounted on a UAV" has been removed to avoid repetition (see Line 83 in Sihoyiya et al., 2025_Track_Changes).

**Commented:** Line 95: varying from ? to ?

which order of magnitude?

We thank the reviewer for this comment. The three manganese-rich layers near the study area are now specified as having variable thicknesses ranging from 5 to 10 m (see Line 115 in Sihoyiya et al., 2025_Track_Changes).

**Commented:** Line 99: again, how thick are these layers? Can we expect that they can be resolved using refleciton seismics?

We thank the reviewer for this comment. In the study area, the manganese-bearing layers are relatively thin, ranging from 5 to 8 m, but can reach up to ~45 m south of the study site.

- Layers in the study area cannot be resolved by reflection seismics because their thickness is below the data's resolution limit.

**Commented:** Line 107 – 118: I think this should not be part of the geology section. You can move this to the seismic section instead.

Thank you for the comment. The section provides important geological context relevant to the seismic interpretation and is therefore briefly introduced here to establish the geological framework. However, we acknowledge your suggestion and will streamline this part in the geology section, while expanding the discussion of its seismic relevance in the seismic interpretation section for better structural coherence.

**Commented:** Line 138: delete high resolution

We thank the reviewer for this comment. The term "high-resolution" has been deleted from the manuscript (see Line 160 in Sihoyiya et al., 2025_Track_Changes).

**Commented:** Line 169: noise

We thank the reviewer for this comment. The word "noise" has been added after "background" to specify background noise (see Line 187 in Sihoyiya et al., 2025_Track_Changes).

**Commented:** Line 163: what is this stripe pattern?

We thank the reviewer for this comment. The stripe patterns observed in the figure were artifacts from the visualization software.

- Corrected image: A version of the figure without the stripe patterns has been added (see Line 190 in Sihoyiya et al., 2025_Track_Changes).

**Commented:** Line 163: Typo : ground roll, guided waves

We thank the reviewer for this comment. The terms "ground roll" and "guided waves" have been corrected (see Line 190 in Sihoyiya et al., 2025_Track_Changes).

**Commented:** Line 182: there is not shot 98 figure 1.

It would also be good to highlight the shown shots in fig. 1. And additionally show shots at the same locations for the crossing lines.

We thank the reviewer for this comment. We considered showing the example shot locations on the map; however, we decided against it because Figure 2 displays only a subset of shot labels for clarity, and including all shots would compromise visualization.

- Example shot gathers added: New example shot gathers at intersection points of crossing lines have been added and corrected (see Line 228 in Sihoyiya et al., 2025_Track_Changes).

**Commented:** Line 195: What is the reason for 10Hz lower boundary?

Thank you for the comment. The lower boundary of 10 Hz was chosen because most of the raw shot gathers are dominated by low-frequency ground roll energy in the 8–14 Hz range. Setting the lower cutoff at 10 Hz was intended to mitigate the influence of ground roll energy.

**Commented:** Line 218: new paragraph.

We thank the reviewer for this comment. A new paragraph has been started at Line 283 to improve readability (see Sihoyiya et al., 2025_Track_Changes).

**Commented:** Line 240: Is the initial model still part of the results? can you restrict the velocity models to the ray-covered parts?!

Thank you for the helpful comment. The tomographic velocity models presented in Figures 7 and 9 are indeed restricted to the ray path–covered zones. However, when shown as standalone models, limiting them strictly to these zones reduces their visual clarity and continuity.

**Commented:** Line 240: It would also be good to compare the intersections of the profiles in more detail, eg using velocity vs. Depth profiles at these locations

We thank the reviewer for this suggestion. A 3D visualization of the refraction tomographic velocity model has been added to better illustrate and compare the intersections of the profiles (see Line 475 in Sihoyiya et al., 2025_Track_Changes).

**Commented:** Line 286: which numbers?

We thank the reviewer for this comment. The numbers now clearly refer to the reflector contacts, and this has been corrected and updated on Line 381 in Sihoyiya et al., 2025_Track_Changes.

**Commented:** Line 308 – 313: move this to 3.3

We thank the reviewer for this comment. The content from Lines 308–313 has been moved to the Refraction Tomography chapter (see Line 419 in Sihoyiya et al., 2025_Track_Changes).

**Commented:** Line 355: how would you optimize a future survey in this region, based on your experience with these data?

We thank the reviewer for this comment. From the experience from this data I would make a recommendation to acquire more data with the following optimizations:

- Increase maximum offset and spread length to enhance velocity analysis and improve depth imaging of deeper targets beneath thick sand cover.
- Incorporate three-component (3C) geophones on selected profiles to record S-wave information, helping discriminate between surface waves and true reflection events.
- Use the current survey results to guide future 3D seismic surveys and subsequent deep exploration boreholes targeting mineralization at depths between 1.0 km and 1.5 km.

The manuscript has been updated on Line 500 of Sihoyiya et al., 2025_Track_Changes.

**Commented:** Line 369 – 375: I cannot agree with your argument. Both, explosives and conventional vibroseis trucks are not cauing problems when operating on sand....

We thank the reviewer for this comment. The statement suggesting that explosives and conventional vibroseis trucks cause problems when operating on sand has been removed (see Lines 529 and 531 in Sihoyiya et al., 2025_Track_Changes).

**Commented:** Line 379: please add examples to demonstrate the repeatability your mentioned enhanced signal-to-noise ratio

We thank the reviewer for this valuable suggestion. While we have not conducted a detailed comparison of the unstacked and stacked shot gathers, the manuscript has been updated to emphasize the processing steps that contributed to the enhanced signal-to-noise ratio and the

repeatability of the results across multiple profiles (see Line 537 in Sihoyiya et al., 2025_Track_Changes).

---

## Author Response (AR3)

Private Bag 3, Wits 2050, Johannesburg, South Africa – Tel: +27 79 577 629 4

Email: 1284606@students.wits.ac.za

**School of Geosciences**

10 December 2025

Dear Prof. Juhlin

We are pleased to submit the revised manuscript eguphere-2025-3117 entitled "**Reflection seismic imaging of the manganese mineralisation in the Griqualand West Basin, South Africa**" for publication. We appreciate and have addressed the constructive comments from the Editor, the Assistant Editor, and the reviewers. Below we reviewed the comments and incorporated them into the manuscript, please note that the original comments are printed in blue for editor, and green for reviewer 2. The line numbers are referred to in the file Sihoyiya et al._track_changes.docx. All the changes to the manuscript are tracked in the file Sihoyiya et al._track_changes.docx. A clean version of the manuscript is also provided in the file Sihoyiya et al._clean.docx.

We believe that the manuscript is now much improved, and we hope that you find it acceptable for publication.

Sincerely,

Mpofana Sihoyiya and co-authors

**Commented:** Figure 4: Change the title of 4b to "Power spectrum", this is what you are showing.

- Thank you for the comment. The title of Figure 4b has been revised to "Power spectrum" to accurately reflect the content shown. This correction has been implemented on Line 180 of Sihoyiya et al. 2025_Track_Changes.

**Commented:** Figure 5: Change the title of 5k to 5o to "Amplitude spectrum", this is what you are showing. Also change the naming of the profiles as suggested by Reviewer #2.

- Thank you for the comment. The titles of Figures 5k to 5o have been revised to "Amplitude spectrum" to accurately reflect the displayed content. In addition, the naming of the profiles has been updated to 'P1 to P5' as suggested by Reviewer #2. These corrections have been implemented on Line 200 of Sihoyiya et al. 2025_Track_Changes.

**Commented:** Table 2, step 3: change 2.5 m binning to (P3-P5), it is currently (P1-P5).

- Thank you for the comment. The binning specification in Table 2, Step 3 has been corrected from (P1–P5) to (P3–P5). This change has been implemented on Line 232 of Sihoyiya et al. (2025_Track_Changes).

**Commented:** I would recommend changing the names of the profiles in Table 1 to the same style as in the text (e.g., P1 instead of MN_P1, etc.). You mention that you did not show P3 due to poor data quality; however, since you also present the data in Figure 5, I would suggest including the corresponding result as well.

- Thank you for the comment. The profile naming in Table 1 has been revised to match the style used in the text (e.g., P1 instead of MN_P1). In addition, although P3 was initially excluded due to poorer data quality, its corresponding result has now been included in Figure 7 to ensure completeness and consistency with Figure 5. This update has been implemented on Line 294 of Sihoyiya et al. 2025_Track_Changes.

**Commented:** Thank you for adding Figure 9. I have an additional question regarding this figure: Discontinuity 1 appears to have no impedance contrast. What is the reason for the reflection observed in your data? Furthermore, can you identify a reverse polarity for reflections 5 and 6 in your image?.

- Thank you for the comment. The impedance contrast for Reflection 1 has been corrected in Figure 9 (Line 374). The polarity reversals observed at Reflections 5 and 6 are interpreted to result from negative acoustic impedance contrasts across these boundaries, caused by a combination of decreasing velocity and/or density with depth and the complex impedance relationship between the Hotazel BIFs and overlying Mooidraai dolomites.

**Commented:** Finally, I would recommend showing the synthetic seismogram converted to depth instead of time. This would make direct comparison with your image easier.

- Thank you for the comment. The synthetic seismogram has now been converted to depth to allow for a more direct and meaningful comparison with the seismic images. The updated depth-converted synthetic seismogram has been added on Line 374 of Sihoyiya et al. 2025_Track_Changes.